# A cell-free nutrient-supplemented perfusate allows four-day ex vivo metabolic preservation of human kidneys

Marlon J. A. de Haan [1,2,10], Marleen E. Jacobs [1,2,10], Franca M. R. Witjas[1,10], Annemarie M. A. de Graaf[1], Elena Sánchez-López [3], Sarantos Kostidis [3], Martin Giera [2,3], Francisco Calderon Novoa [4], Tunpang Chu[4], Markus Selzner[4], Mehdi Maanaoui[5], Dorottya K. de Vries[6,7], Jesper Kers[8,9], Ian P. J. Alwayn[6,7], Cees van Kooten[1], Bram Heijs[2,3], Gangqi Wang[1,2,11] ✉, Marten A. Engelse[1,2,11] ✉ & Ton J. Rabelink [1,2,11] ✉

The growing disparity between the demand for transplants and the available donor supply, coupled with an aging donor population and increasing prevalence of chronic diseases, highlights the urgent need for the development of platforms enabling reconditioning, repair, and regeneration of deceased donor organs. This necessitates the ability to preserve metabolically active kidneys ex vivo for days. However, current kidney normothermic machine perfusion (NMP) approaches allow metabolic preservation only for hours. Here we show that human kidneys discarded for transplantation can be preserved in a metabolically active state up to 4 days when perfused with a cell-free perfusate supplemented with TCA cycle intermediates at sub-normothermia (25 °C). Using spatially resolved isotope tracing we demonstrate preserved metabolic fluxes in the kidney microenvironment up to Day 4 of perfusion. Beyond Day 4, significant changes were observed in renal cell populations through spatial lipidomics, and increases in injury markers such as LDH, NGAL and oxidized lipids. Finally, we demonstrate that perfused kidneys maintain functional parameters up to Day 4. Collectively, these findings provide evidence that this approach enables metabolic and functional preservation of human kidneys over multiple days, establishing a solid foundation for future clinical investigations.

Numerous studies have demonstrated that machine perfusion surpasses static cold storage (SCS) in terms of transplantation outcomes, thereby enabling the transplantation of organs that would otherwise be discarded[1–5]. Consequently, there has been a noticeable shift away from SCS-based preservation towards perfusion-based preservation methods. With technological advancements and the development of more sophisticated perfusion platforms, there is a growing awareness of the benefits of maintaining perfused organs in a metabolically active

state, known as warm perfusion. This approach holds tremendous potential for resuscitation, repair, rejuvenation, and even regeneration of deceased donor organs[6,7].

While significant progress has been made in extending perfusion over multiple days for liver transplantation[8–10], unfortunately, this approach cannot be readily applied to the kidney, which constitutes the majority of transplant organs. The extracorporeal perfusion of red blood cell (RBC)-based perfusates through the kidney invariably leads

to haemolysis, resulting in the accumulation of free haemoglobin. Unlike the liver, which can convert free haemoglobin into bilirubin and excrete it into bile, in the kidney, this process directly leads to acute kidney injury[11,12]. RBC-based perfusion of human kidneys with subsequent transplantation has only been described up to 6 h, and even then, there is no definitive proof of functionality[13]. Moreover, the kidney has an exceptionally high metabolic demand to maintain electrolyte reabsorption and corticomedullary gradients, making metabolic preservation a critical objective, especially for kidney donation[14].

Two decades ago, Brasile et al. proposed cell-free perfusion at subnormothermia as approach for prolonged ex vivo preservation of donor kidneys[15–17]. Lowering preservation temperature reduces the metabolic rate, and therewith oxygen requirements. It has been demonstrated that at subnormothermia (20–32 °C), the oxygen demand of the graft can be met using a cell-free perfusate, thereby negating the need for RBCs[18–21]. Moreover, it was recently demonstrated that at subnormothermia (28 °C) there is sufficient metabolic activity to support molecular and cellular repair processes[22].

In this study, we perfused human kidneys discarded for transplantation with a cell-free perfusate supplemented with TCA cycle intermediates at subnormothermia (25 °C) up to 8 days. Spatial metabolic flux measurements, a method we recently developed[23,24], were performed on kidney biopsies taken at different timepoints during perfusion. Through this approach, we were able to dissect the specific nutrients required for the metabolic preservation of the kidney. Our findings demonstrate that by using a cell-free nutrient-supplemented perfusate, we can achieve both metabolic and structural/functional preservation of discarded human donor kidneys for at least 4 days.

## Results

### Subnormothermic culture platform

The primary objective of this study was the development of a platform that supports multi-day ex vivo preservation of metabolically active human kidneys. The platform consists of a custom made air-tight organ chamber (Fig. 1A) within which continuous assessment of hemodynamic parameters and perfusate sampling through arterial, venous, and ureteral cannulation is possible. Renal flow is maintained by a pressure controlled centrifugal pump set at 75 mmHg. The kidneys are perfused with an acellular perfusate containing DMEM F12 and human serum albumin as main components. To support renal metabolism we utilized a perfusate rich in glutamine, citrate and acetate (for full composition see Supplementary Table 1)[24–27]. The temperature was kept constant at 25 °C throughout the entire culture period, and oxygenation was achieved by saturating the acellular perfusate with carbogen (95% $O_2$, 5% $CO_2$) using a long-term oxygenator. Initial experiments using porcine kidneys have demonstrated the effectiveness of this method in delivering oxygen at subnormothermia (Supplementary Fig. 1).

To optimize hemodynamic control and electrolyte balance, urine was recirculated, as previous studies have shown its beneficial effects[28–30]. To address the accumulation of metabolic waste products in a closed system during prolonged perfusion, we incorporated continuous haemodialysis. Continuous haemodialysis is driven by an independent pump, which removes small molecular waste products through a dialysis filter at a rate of 40 mL/h. Substitution solution (for composition see Supplementary Table 1) is added post-filter at the same rate, resulting in a continuous 5% fluid exchange.

### Perfusion dynamics during 8-day culture

We perfused three human kidneys that were deemed unsuitable for transplantation (for donor data see Supplementary Table 2). The main objective in this phase was to explore the potential duration of ex vivo preservation using this platform, and to describe what happens to a kidney graft as it fails during long-term perfusion. The perfusion of all

three kidneys was ended after an 8-day period due to progressive disturbances in perfusion dynamics (Fig. 1).

Upon connection of the kidneys to the culture platform, renal flow increased during the first hours of perfusion, accompanied by a decrease in vascular resistance (Fig. 1B, C). Renal flow remained stable until Day 5-Day 6 of perfusion, after which it gradually decreased. Oxygen delivery has a linear relationship with renal flow and remained between 5 and 15 mL $O_2$/min throughout the 8-day period (Fig. 1D). Oxygen uptake was constant during the first days and lactate clearance was observed. However, after Day 4, oxygen uptake decreased below 0.5 ml $O_2$/ min (Fig. 1D). Concurrently, the perfusate pH for all three kidneys decreased below 7.30 (Fig. 1E), while there was a progressive increase in perfusate lactate leading up to Day 8 (Fig. 1F). Perfusate glucose remained within normal range (Fig. 1G). These observations suggest a metabolic shift towards glycolysis.

The perfusate level of lactate dehydrogenase (LDH), an enzyme marker for cellular injury, indicated relatively low cellular injury until Day 4 of perfusion, followed by a progressive increase up to Day 8 (Fig. 1H). Tubular injury markers KIM1 and NGAL were measured within the perfusate as urine was recirculated (Fig. 1I, J). Perfusate KIM1 exhibited a gradual increase throughout the 8-day period, whereas perfusate NGAL levels demonstrated the largest increase between Day 6 and Day 8. The potassium concentration increased following the start of perfusion but remained within normal range thereafter for the 8-day period (Fig. 1K). By Day 8, kidney weight had increased with 28 ± 1% compared to baseline (Fig. 1L; Day 0: 283 ± 39, Day 8: 363 ± 52 g). Throughout the 8-day period there were no noticeable changes in the macroscopic appearance of the kidneys (Fig. 1M). Histological analysis of tissue biopsies taken during perfusion revealed progressive damage to the apical membrane of tubular cells, including the loss of tubular lumen between Day 6 and Day 8 (Fig. 1N).

### Lipidomics at single cell resolution highlight phenotypic remodeling

To gain a deeper understanding of lipidomic and metabolomic changes during multi-day kidney perfusion, cortex biopsies (4 mm) were taken at five different timepoints (Day 0, Day 2, Day 4, Day 6, Day 8) throughout the 8-day perfusion period from all three human kidneys. These biopsies were subjected to high spatial resolution matrix-assisted laser desorption/ionization mass spectrometry imaging (MALDI-MSI), as previously described[24]. This technique allows the simultaneous mapping of hundreds of lipids and metabolites per pixel while providing spatial information at subcellular resolution ($5 \times 5 \ \mu m^2$ pixel size, with an average renal cell diameter of approximately 10 μm)[31]. Following MALDI-MSI measurements, we performed Uniform Manifold Approximation and Projection (UMAP) analysis based on the lipidomic data, combined with post-MALDI-MSI immunofluorescence staining for cell type identification (Fig. S2A). We could distinguish LTL+ proximal tubules, ECAD+ tubules and podocytes from other renal cell types (Fig. 2A and Supplementary Fig. 2) based on distinct lipid signatures, composed of phospholipids predominantly present in cell membranes. Proximal tubule, ECAD+ tubule and podocyte identity was confirmed by immunofluorescence staining on post-MALDI-MSI analysed tissue through alignment with cell type distribution (Supplementary Fig. 2A, B).

First, we evaluated the impact of 8-day ex vivo perfusion on cell phenotypic preservation by analysing changes in lipid species profiles. Unsupervised clustering was performed for the ECAD+ tubules (Fig. 2B), podocytes (Fig. 2C) and proximal tubules (PT) (Fig. 2D), respectively. Different subsets of cell phenotypes were observed, characterized by distinct lipid signatures (Fig. 2B–D). Throughout the 8-day perfusion period, a gradual shift in relative composition of the different cell phenotypes occurred. Unsupervised hierarchical clustering of the different timepoints for each cell population revealed that lipid remodeling predominantly occurred between Day 6 and Day 8 of

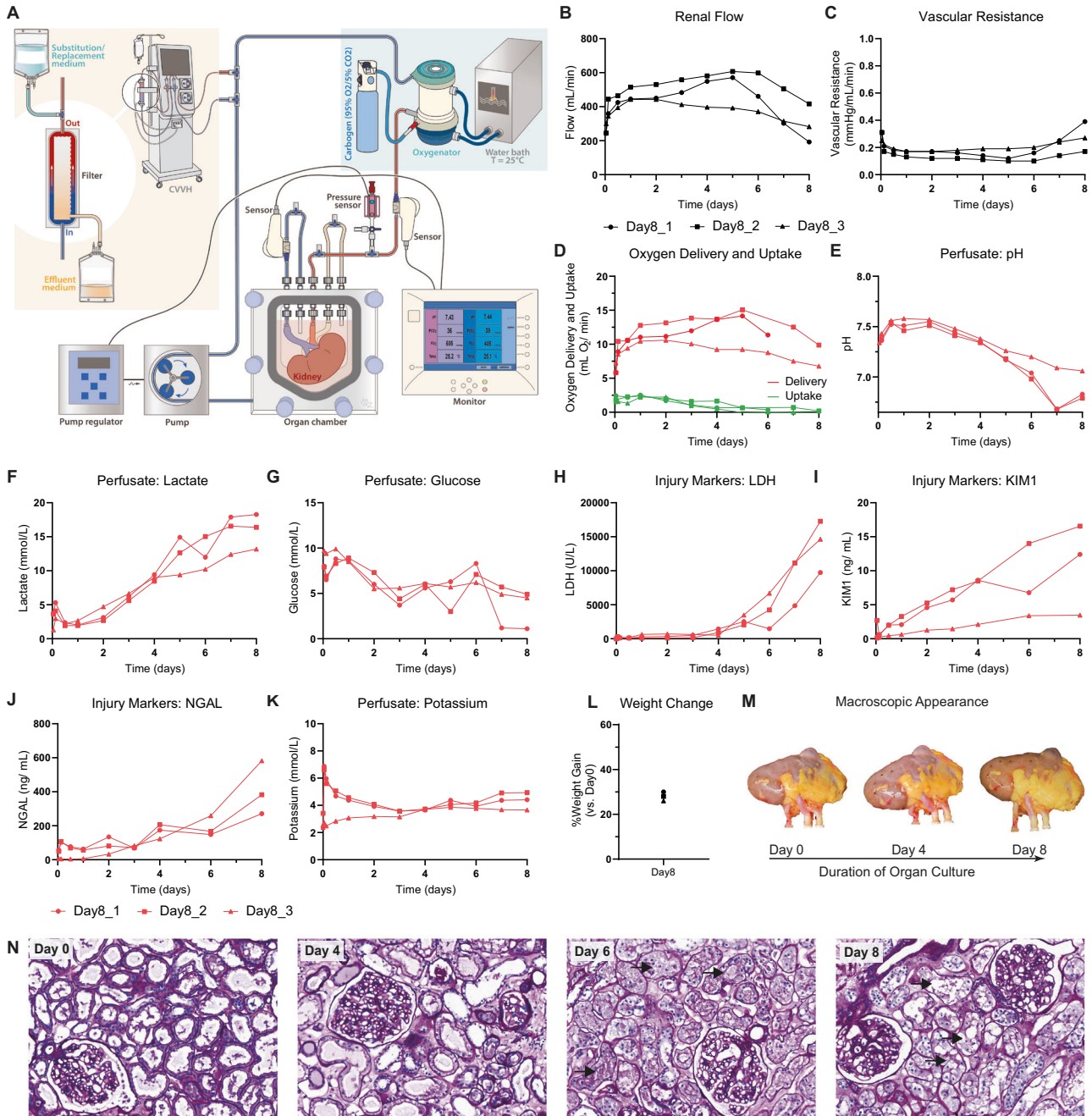

**Fig. 1 | Perfusion dynamics during 8-day subnormothermic culture of human kidneys.** Three discarded human kidneys were cultured for an 8-day period. **A** Schematic overview of organ culture platform. Kidneys were preserved in a custom-designed organ chamber. A pressure controlled centrifugal pump set at 75 mmHg perfused the renal artery with acellular perfusate after it passed through an oxygenator. Perfusion temperature was maintained at subnormothermia (25 °C) throughout the entire culture period. Arterial, venous and ureteral cannulation allowed continuous assessment of perfusion parameters. Urine was recirculated. Continuous hemofiltration allowed removal of small molecular weight waste products and substitution with fresh solution. Renal flow (**B**) and vascular resistance (**C**) throughout the 8-day period. **D**–**G** Metabolic perfusate dynamics throughout the 8-day period. Oxygen delivery as calculated from the pO$_2$ in the arterial inflow and oxygen uptake as calculated from the delta pO$_2$ between the arterial inflow and

venous outflow (**D**). Perfusate pH (**E**), perfusate lactate (**F**), and perfusate glucose (**G**) as measured in the arterial inflow. **H**–**K**, Perfusate injury markers throughout the 8-day period. Perfusate LDH (**H**) as marker for general cell damage. Perfusate KIM1 (**I**) and perfusate NGAL (**J**) as markers for proximal tubular and distal tubular injury, respectively. Because urine is recirculated tubular injury markers were measured within the perfusate (arterial inflow). Perfusate potassium as marker for general cell damage (**K**). **L** Weight gain (%) after the 8-day period. **M** Macroscopic appearance of discarded human kidneys throughout the 8-day period. One out of three representative kidneys shown. **N** Representative images of PAS histology on cortex biopsies (*n* = 3 perfusions with biopsies taken at Day 0, Day 2, Day 4, Day 6 and Day 8 from each kidney). Black arrows highlight areas that demonstrate loss of tubular lumen. Bars represent 100 μm. Source data are provided as a Source Data file.

perfusion (Fig. 2E, G, I). For example, phenotype PT_3, which was minimally present on Day 0, accounted for more than 30% of PTs on Day 8 and was characterized by a lipid marker at *m/z* 865.6. This marker has previously been identified as an indicator of injury in proximal

tubules following ischemia/reperfusion injury in a murine model[24]. In parallel, a decrease of PT_1 and PT_2 phenotypes was observed from Day 0 to Day 8. Comparing the spatial segmentation with immunofluorescence staining revealed that the abnormal PT_3 phenotype

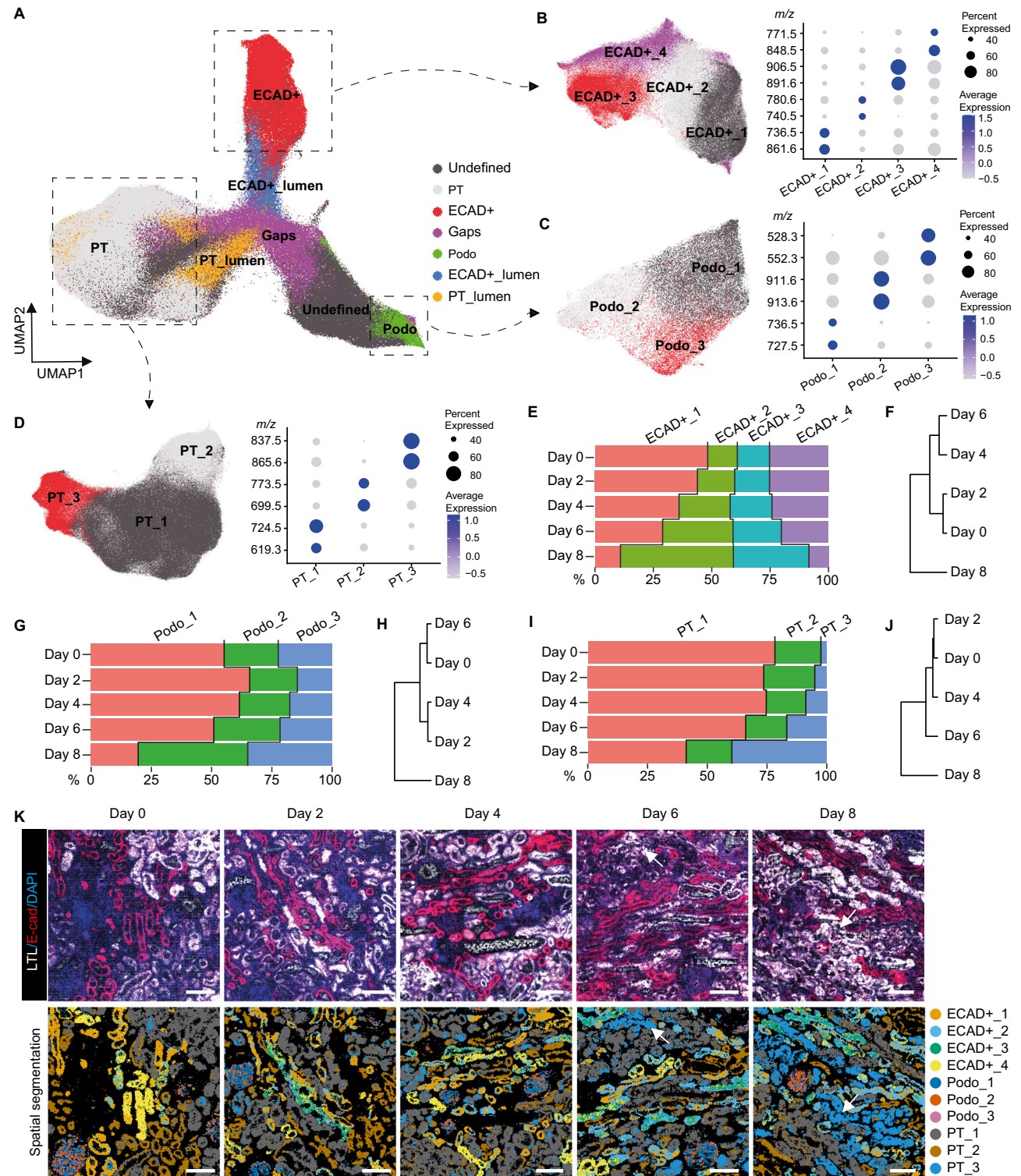

**Fig. 2 | Spatial lipidomics reveals epithelial phenotypic remodeling during 8-day subnormothermic culture of human kidneys. A** Lipid heterogeneity in biopsies taken at different timepoints (Day 0, Day 2, Day 4, Day 6, Day 8) from human kidneys ($n = 3$) throughout the 8-day culture period allows identification of the main epithelial cell types, visualized in UMAP plot of MALDI-MSI data ($5 \times 5\,\mu m^2$ pixel size). Sub clustering of ECAD+ tubular cells (**B**), podocytes (**C**), and proximal tubular cells (PT) (**D**) displays different epithelial phenotypes within each cell type. Dot plots display expression of cluster-specific lipid features. **E** Percentage of each ECAD+ tubular phenotype in biopsies taken at different timepoints during 8-day culture. **F** Hierarchical clustering of the percentage of ECAD+ tubular phenotypes in biopsies taken at different timepoints. **G** Percentage of each podocyte phenotype

in biopsies taken at different timepoints during 8-day culture. **H** Hierarchical clustering of the percentage of podocyte phenotypes in biopsies taken at different timepoints. **I** Percentage of each tubular PT phenotype in biopsies taken at different timepoints during 8-day culture. **J** Hierarchical clustering of the percentage of PT tubular phenotypes in biopsies taken at different timepoints.
**K** Immunofluorescence staining on post-MALDI-MSI samples taken at different timepoints during 8-day organ culture (top panel). Spatial segmentation showing distribution of different renal epithelial phenotypes (bottom panel) ($n = 3$ perfusions with biopsies taken at Day 0, Day 2, Day 4, Day 6 and Day 8 from each kidney). White arrows highlight the area of PT_3. Bars represent 200 μm. Source data are provided as a Source Data file.

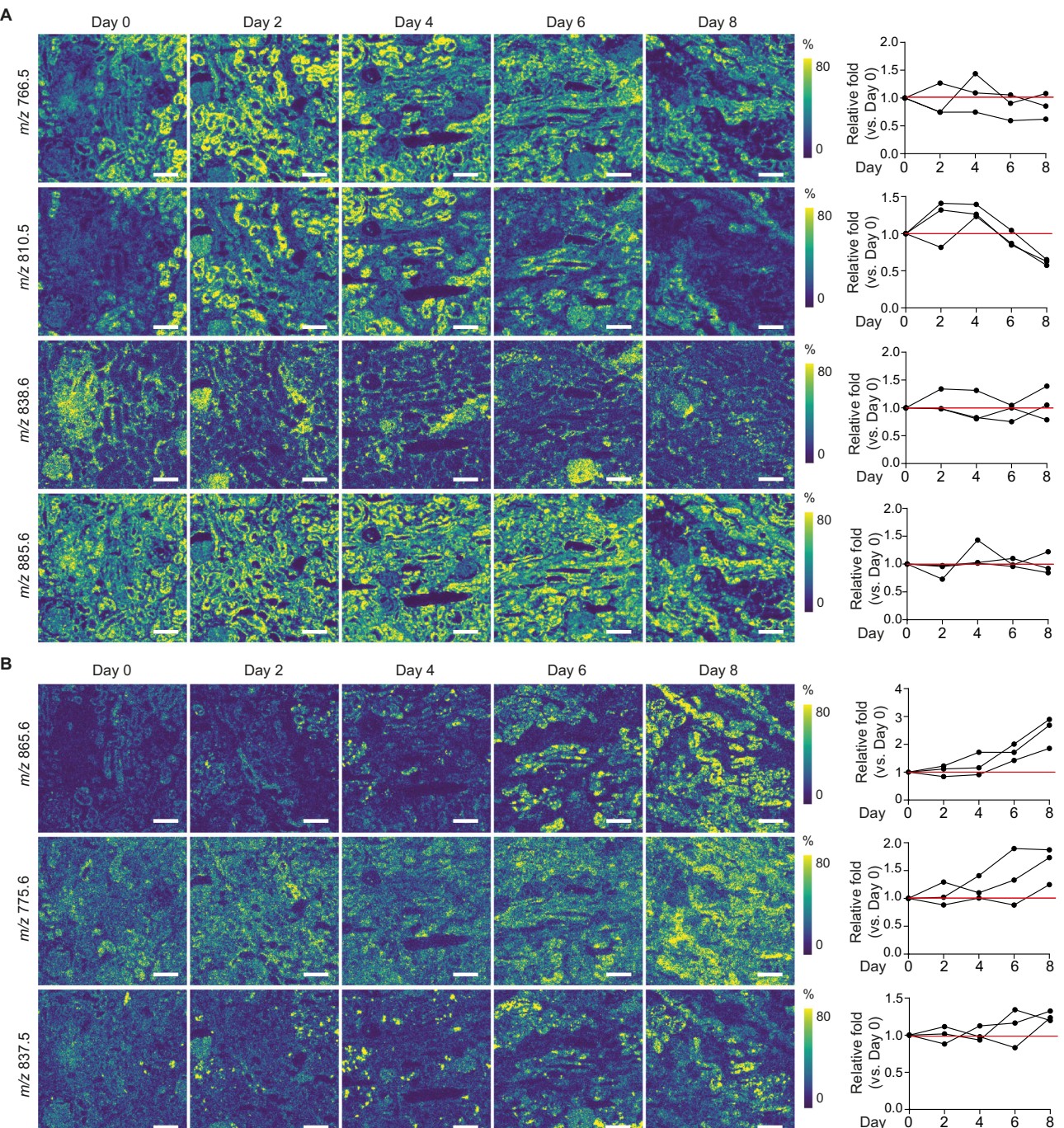

**Fig. 3 | Spatial lipid species distribution changes during 8-day sub-normothermic culture of human kidneys. A** Representative images demonstrate changes in spatial distribution of lipid species characteristic for normal renal cell phenotypes at different timepoints during 8-day organ culture, as recorded by MALDI-MSI (5 × 5 μm² pixel size) (*n* = 3 perfusions with biopsies taken at Day 0, Day 2, Day 4, Day 6 and Day 8 from each kidney). Lipid species *m/z* values are representative for PT_1 and PT_2 (*m/z* 766.5), PT_1 (*m/z* 810.5), podocytes (*m/z* 838.6), and all tubular structures (*m/z* 885.6). Right panel shows relative fold change compared to baseline (Day 0) for each timepoint during culture. Scale bar = 200 μm. **B** Representative images demonstrate changes in spatial distribution of lipid species that are characteristic for abnormal PT phenotype (PT_3) at different timepoints during 8-day organ culture, as recorded by MALDI-MSI (5 × 5 μm² pixel size) (*n* = 3 perfusions with biopsies taken at Day 0, Day 2, Day 4, Day 6 and Day 8 from each kidney). Right panel shows relative fold change compared to baseline (Day 0) for each timepoint during culture. Bars represent 200 μm. All images generated from same samples as shown in Fig. 2K with IF staining and spatial segmentation. Bars represent 200 μm. Source data are provided as a Source Data file.

mainly colocalized with LTL staining in the Day 6 and Day 8 biopsies (Fig. 2K), confirming the phenotypic changes within proximal tubules. These observations suggest that cellular phenotypes are preserved up to Day 4-Day 6 of ex vivo preservation.

To further assess how the relative abundances of lipids change during the 8-day culture, we compared the lipid species characteristic for the main tubular subsets (Fig. 3A, B). Due to donor variation, average lipid peak intensities were normalized to their level at Day 0. The spatial distributions and relative abundances of the different lipid markers for the main renal cell types remained similar until Day 6 of perfusion (Fig. 3A). This suggests preservation of the main cell membrane lipid components without degradation up to Day 6-Day 8 of

ex vivo perfusion. In contrast, the relative abundance of lipid markers characteristic for the PT_3 proximal tubules exhibited a considerable increase on Day 8 as compared to Day 0 (Fig. 3B). Assessment of their spatial distribution and relative abundance indicates that this remodeling began between Day 4 and Day 6, progressively worsening towards Day 8, as evidenced by a more homogenous distribution throughout the tissue.

To investigate the underlying factors contributing to the changes in lipid species profile, we assessed the occurrence of lipid peroxidation. From Day 6 of perfusion onwards, we observed a progressive increase in the perfusate levels of malondialdehyde (MDA) (Fig. 4A), an indicator of oxidative stress. Targeted lipidomics confirmed the increased abundance of oxidized lipids in the perfusate on Day 8 as compared to Day 0 through Day 4 (Supplementary Fig. 3). While the abundance of oxidized lipids in the perfusate was relatively low until Day 4 to Day 6 of perfusion, a significant increase was observed on Day 8.

Next, we performed untargeted lipidomics analysis on kidney tissue perfused for 8-days, comparing it with non-perfused kidney control tissue. In Day 8 tissue, the lipid species demonstrating the most significant fold change were all oxidized phospholipid species (Fig. 4B). The spatial distribution of these oxidized phospholipids was assessed within the MALDI-MSI analysed biopsies (Fig. 4C). This supports that oxidized lipid species are absent between Day 0 and Day 4-Day 6 of ex vivo perfusion, with progressive oxidative stress taking place between Day 4-Day 6 and Day 8 of perfusion resulting in the accumulation of peroxidised lipid species towards Day 8. Additionally, comparing the spatial distribution of the oxidized phospholipid PE(O-36:2) on Day 8 of perfusion (Fig. 4C) with the spatial segmentation of

the different cell clusters (Fig. 2K) demonstrates that it co-localizes with the PT_3 proximal tubules. These findings indicate that oxidative stress progressively worsens after Day 4 of perfusion. Consequently, we proceeded to evaluate the metabolic activity using spatially resolved isotope tracing after 4- and 8-day ex vivo perfusion.

## Isotope tracing indicates preserved metabolic activity during prolonged culture

Defective metabolism and metabolic recovery play a key role in the response to kidney injury[32,33]. To assess preservation of central carbon metabolism and changes in nutrient partitioning during 8-day perfusion we applied our recently described spatial dynamic metabolomics platform[24]. In short, dedicated cortex biopsies were taken on Day 0, Day 4 and Day 8 of perfusion and incubated with $^{13}C$-labeled glucose or glutamine for 2 h at 37 °C. Next, we used MALDI-MSI to spatially visualize $^{13}C$-enrichment of downstream intermediates of glycolysis and the TCA cycle at subcellular resolution.

We found that all renal cell types identified in the biopsies maintained active cell metabolism and could use glucose and glutamine as nutrient sources for glycolysis and the TCA cycle after 4- and 8-days of perfusion (Fig. 5). The enrichment of the $^{13}C_6$-glucose-derived isotopologues, $^{13}C_3$–3PG, $^{13}C_3$-Lactate and $^{13}C_2$-Glutamate demonstrated active glycolysis and contribution of glucose to the TCA cycle, respectively (Fig. 5A–F). The enrichment of $^{13}C_5$-glutamine-derived isotopologues, $^{13}C_5$-Glutamate, $^{13}C_4$-Succinate, $^{13}C_4$-Malate and $^{13}C_3$-Glutamate, demonstrated passage through the entire oxidative TCA cycle (Fig. 5H–N). A partial contribution of glutamine to reductive TCA could not be ruled out.

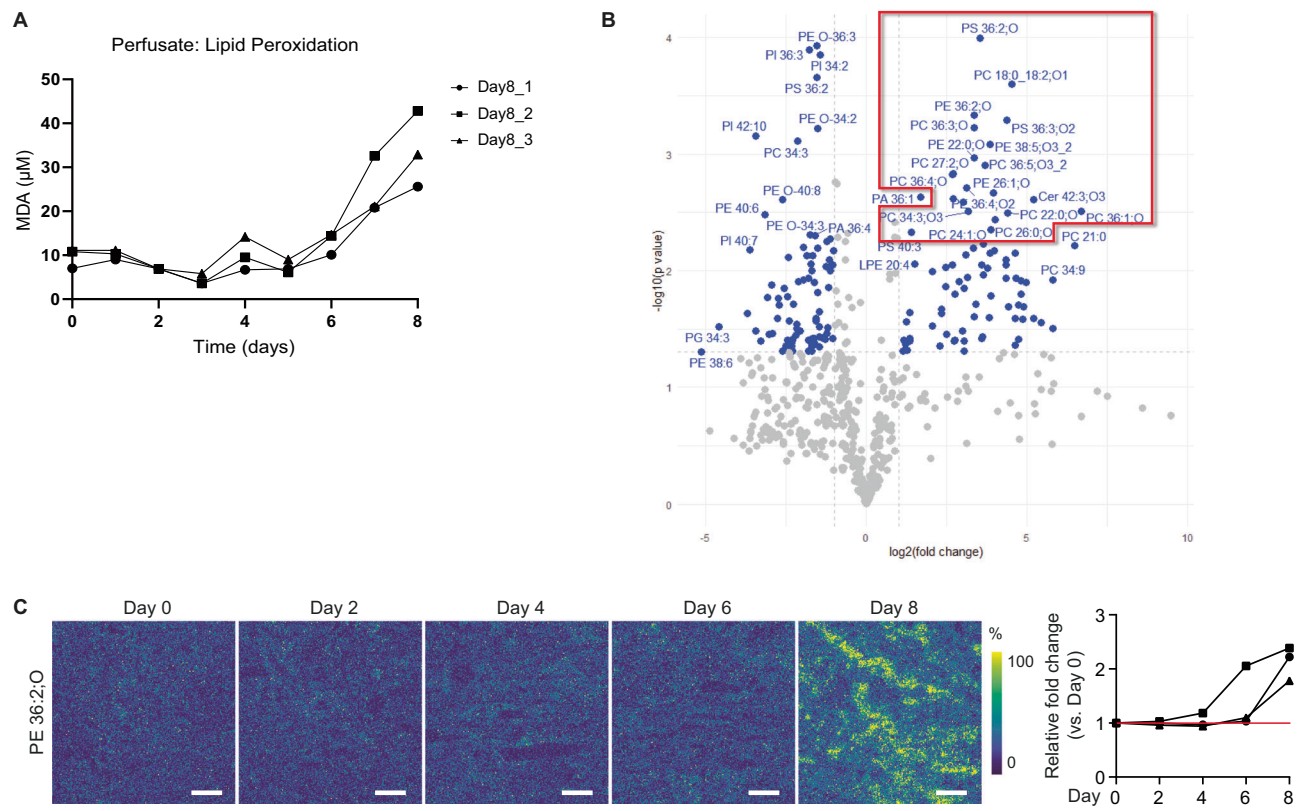

**Fig. 4 | Oxidative damage during 8-day subnormothermic culture of human kidneys. A** Perfusate malondialdehyde (MDA) as marker for lipid peroxidation during 8-day perfusion. **B** Volcano plot demonstrating log2 fold change in lipid species measured in 8-day perfused tissue samples as compared to control kidney tissue ($n = 3$ human kidneys per group), as identified through untargeted lipidomics. Two-sided $t$ test. **C** Representative images demonstrate changes in spatial distribution of oxidized phospholipid species (PE 36:2;O) at different timepoints during 8-day perfusion, as recorded by MALDI-MSI ($5 \times 5 \mu m^2$ pixel size) ($n = 3$ perfusions with biopsies taken at Day 0, Day 2, Day 4, Day 6 and Day 8 from each kidney). Right panel shows relative fold change compared to Day 0. Bars represent 200 μm. Source data are provided as a Source Data file.

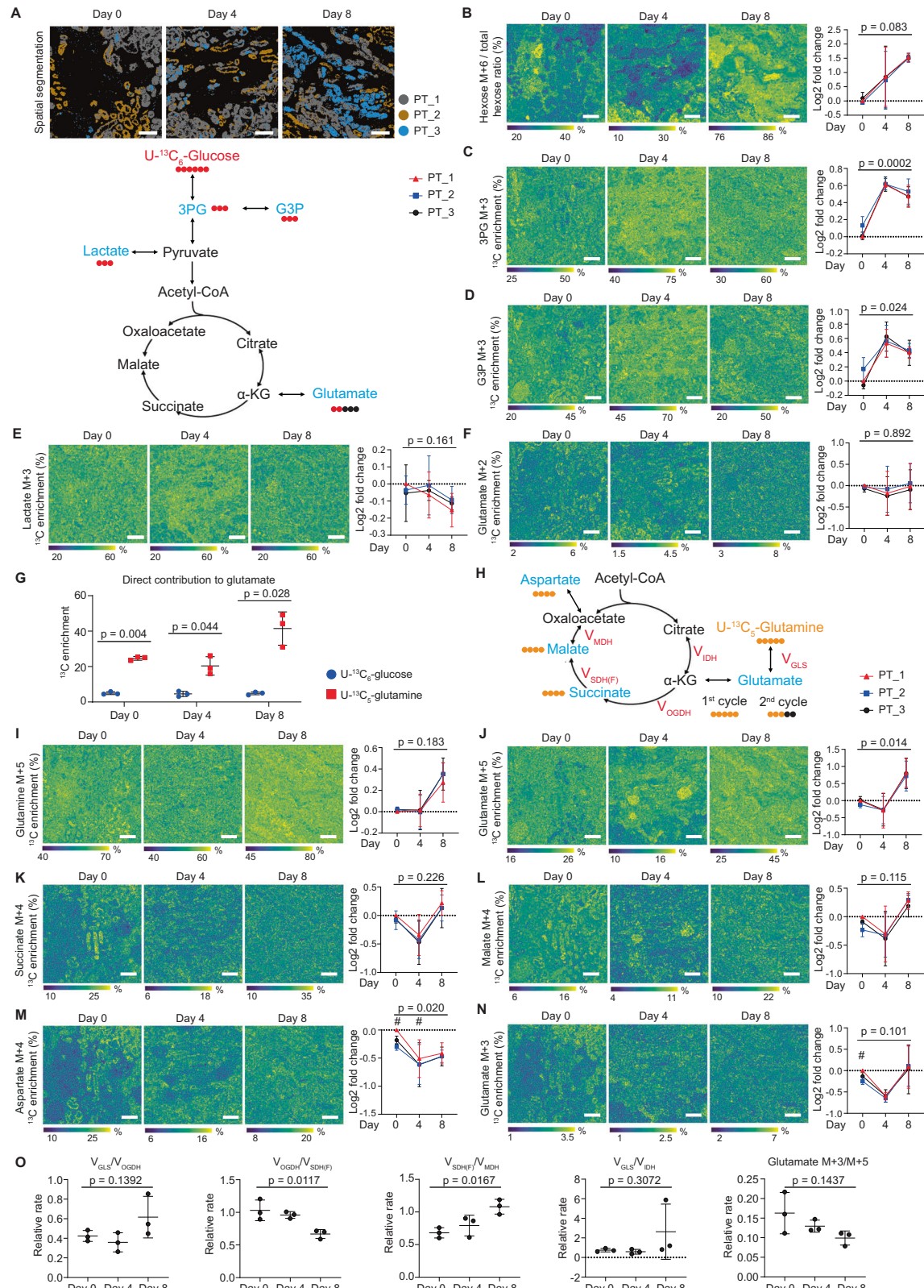

Next, we focused on metabolic changes in the different proximal tubule phenotypes identified during 8-day perfusion (Fig. 5A). Analysis of the isotopologues derived from $^{13}C_6$-glucose revealed that all PT phenotypes exhibited an increase in U-$^{13}C_6$-hexose and $^{13}C_3$–3PG enrichment during perfusion as compared to PT at Day 0 (Fig. 5B, C). A higher $^{13}C_3$-G3P enrichment was also observed after perfusion, pointing to a side branch of glycolysis (Fig. 5D). The enrichment of $^{13}C_3$-

lactate and $^{13}C_2$-glutamate from glucose carbons revealed no differences during 8-day perfusion (Fig. 5E, F). These findings suggest an active glycolysis pathway and the contribution of glucose to the TCA cycle until Day 8 of perfusion. Notably, the carbon contribution of $^{13}C_5$-glutamine to glutamate is about 5 times higher than $^{13}C_6$-glucose (Fig. 5G). As the conversion of glutamate to α-ketoglutarate (α-KG) is very significant in mitochondria, this demonstrates a higher

**Fig. 5 | Spatial dynamic metabolic measurements on biopsies taken at different timepoints during 8-day culture. A** Representative spatial segmentation of proximal tubule phenotypes at different timepoints during 8-day human kidney perfusion ($n = 3$ perfusions with biopsies taken at Day 0, Day 4 and Day 8 from each kidney). Bars represent 200 µm. **B–F** Images and graphs showing the spatial dynamic metabolic measurements using U-$^{13}C_6$-glucose on biopsies after 2 h' incubation ($n = 3$ perfusions with biopsies taken at Day 0, Day 4 and Day 8 from each kidney). Graphs are shown as log2 fold change compared to PT1 at Day 0. Bars represent 200 µm. One way ANOVA test. **G** Direct carbon contribution of different nutrients to glutamate in proximal tubule cells, as measured from the glutamate isotopologues M + 2, M + 3, and M + 5 ($n = 3$ perfusions with biopsies taken at Day 0, Day 4 and Day 8 from each kidney). Two-tailed unpaired $t$ test. **H–N**, Images and

graphs showing the spatial dynamic metabolic measurements using U-$^{13}C_5$-glutamine on the biopsies obtained at different timepoints during the 8-day organ culture period after 2 h incubation ($n = 3$ perfusions with biopsies taken at Day 0, Day 4 and Day 8 from each kidney). Graphs are shown as log2 fold change compared to PT1 at Day 0. Bars represent 200 µm. One way ANOVA test (exact $p$ values in figure). Two-tailed paired $t$ test compared to PT1 (#p < 0.05). **O** Relative flux rate of different TCA cycle steps ($n = 3$ perfusions with biopsies taken at Day 0, Day 4 and Day 8 from each kidney). One way ANOVA test. Data are represented as mean ± SD. All images generated from same samples as shown in Fig. 2K with IF staining, molecular histology and spatial segmentation. Source data are provided as a Source Data file.

contribution of glutamine as carbon source for the TCA cycle as compared to glucose.

The presence of $^{13}$C-labeled TCA cycle intermediates derived from $^{13}C_5$-glutamine confirms the continued activity of the TCA cycle until Day 8 of perfusion (Fig. 5H−N). At Day 0, PT_1 and PT_2 (PT_3 being almost absent) displayed heterogeneity in their metabolic dynamics, as shown by $^{13}C_5$-glutamine-derived $^{13}C_4$-aspartate and $^{13}C_3$-Glutamate (Fig. 5M, N). However, this metabolic heterogeneity between different PT phenotypes was lost during perfusion, particularly at Day 8 (Fig. 5M, N). Comparing Day 8 to Day 0, there was a significant increase in $^{13}C_5$-glutamate and decrease in $^{13}C_4$-aspartate in PT (Fig. 5J, M).

To evaluate the relative enzyme activity in the TCA cycle at different timepoints during perfusion, we conducted Q-Flux analysis ($^{13}C_5$-glutamine-based method) using equations from a recent publication that validated this approach[34]. Previously we demonstrated that after 2 h of ex vivo incubation with $^{13}C_5$-glutamine a pseudo-steady state is reached, allowing Q-flux analysis[24]. This analysis reveals a decrease in the fractional contribution of succinate-to-succinate dehydrogenase forward flux on Day 8, while the succinate dehydrogenase forward flux relative to malate dehydrogenase flux was increased at Day 8 (Fig. 5O). Overall, we observed a decreasing trend in the fraction of $^{13}C_3$-glutamate that is derived from $^{13}C_5$-glutamate through the entire oxidative TCA cycle (Fig. 5O). Most importantly, we observed no differences in the relative rate of TCA cycle enzymes between Day 0 and Day 4 of perfusion (Fig. 5O). This finding underpins that subnormothermic machine perfusion (sNMP) allows multi-day ex vivo metabolic preservation of diseased human donor kidneys.

## Renal function during 4-day culture
Following the demonstration of metabolic preservation for more than 4 days, we included an additional five human kidneys (for donor data see Supplementary Table 2). These kidneys were cultured for a 4-day period during which perfusion dynamics, renal function and tissue histology were assessed (Fig. 6). Renal flow and vascular resistance remained stable throughout the 4-day period (Fig. 6A, B). The kidneys maintained a consistent oxygen uptake around 1 mL $O_2$/min (Fig. 6C, D). The pH levels were within normal range after 3 h of perfusion and ended at 7.37 ± 0.05 on Day 4 (Fig. 6E). In contrast to the observed accumulation of injury markers towards the end of the 8-day perfusions (Fig. 2), relatively minor increases in perfusate LDH, KIM1 and NGAL were observed throughout the 4-day period (Fig. 6H−J). Histological scoring of periodic acid-Schiff (PAS) staining by a renal pathologist was performed on the five kidneys that were perfused for the 4-day period (Fig. 6K, Supplementary Fig. 4). Wedge biopsies were taken at the end of the 4-day perfusion and control biopsies were taken from the contralateral kidneys (only for Day4_1 and Day4_2).

Finally, we assessed renal function during ex vivo preservation. In all kidneys glomerular filtration was observed, as indicated by urine production (Fig. 6L). To examine the integrity of the glomerular filtration barrier, fluorescently labeled dextrans of 20 kDa and 500 kDa were added to the perfusate on Day 1 and Day 3 of perfusion. The 20 kDa dextrans were filtered into the urine whereas the 500 kDa

dextrans were retained within the vascular compartment, indicating preserved barrier function (Fig. 6M, N). During the 4-day sub-normothermic culture period we observed active tubular transport, demonstrated by the reabsorption and secretion of electrolytes and metabolites (Fig. 6O, P). Sodium and glucose were reabsorbed (Fig. 6O) whereas potassium and urea were secreted (Fig. 6P). In contrast, during the 8-day perfusions, the kidneys lost their ability to create electrolyte gradients between Day 4 and Day 6 of preservation (Supplementary Fig. 5).

## Porcine perfusion dynamics do not reflect human perfusion dynamics
To provide proof-of-concept, and to demonstrate that it is feasible to procure, preserve for 4 days, and transplant a kidney, living-donor porcine kidney perfusions and auto-transplantation experiments were performed (Supplementary Figs. 6, 7, $n = 3$). Male Yorkshire pigs (30 kg, 3-month-old) were used for this purpose. Graft retrieval and auto-transplantation were performed as previously described[35,36].

Upon connection of the porcine kidneys to the culture platform, vascular resistance decreased during the first hours of perfusion, accompanied by an increase in renal flow (Supplementary Fig. 6A, B). However, after 12−24 h vascular resistance began to increase, leading to a gradual reduction in renal flow. At this point, perfusion dynamics of these porcine kidneys started to differ considerably from our previous observations in human kidneys (Supplementary Fig. 6E, F). Furthermore, at the conclusion of multi-day perfusion, the porcine kidneys had gained 114−308% compared to their initial weight, a notable contrast to our findings in human kidneys (Supplementary Fig. 6C, D, G).

Because of these distinct differences in perfusion dynamics, we conclude that the outcomes of porcine auto-transplantation following multi-day sNMP will not accurately predict the outcomes of human kidney transplantation. Despite these differences with the human perfused kidneys, two of the perfused porcine kidneys were auto-transplanted after 4-day perfusion (Supplementary Fig. 7). Kidney#1 was transplanted following a contralateral nephrectomy. Initially, a uniformly perfused pink kidney was observed after reperfusion. However, in the following hour the kidney developed haemorrhagic infarction and the pig entered a distributive shock that required large doses of pressors and fluids. The experiment was terminated 2.5 h after reperfusion. We believe the intraparenchymal blood was a reflection of disturbed vascular permeability, as was also supported by the weight gain of the kidney during perfusion. For the transplantation of Kidney#3 the contralateral kidney remained in situ. After reperfusion, a patchy appearance with pink and darker regions was noted, as is often observed in donation after circulatory death. Notably, reperfusion differed significantly from Kidney#1, as minimal pressors and fluids were required to maintain blood pressure. The pig was sacrificed after 7 days. Again, the kidney showed regions of haemorrhagic infarction and intraparenchymal blood, reflective of disrupted vascular permeability, albeit to a lesser extent when compared to Kidney#1.

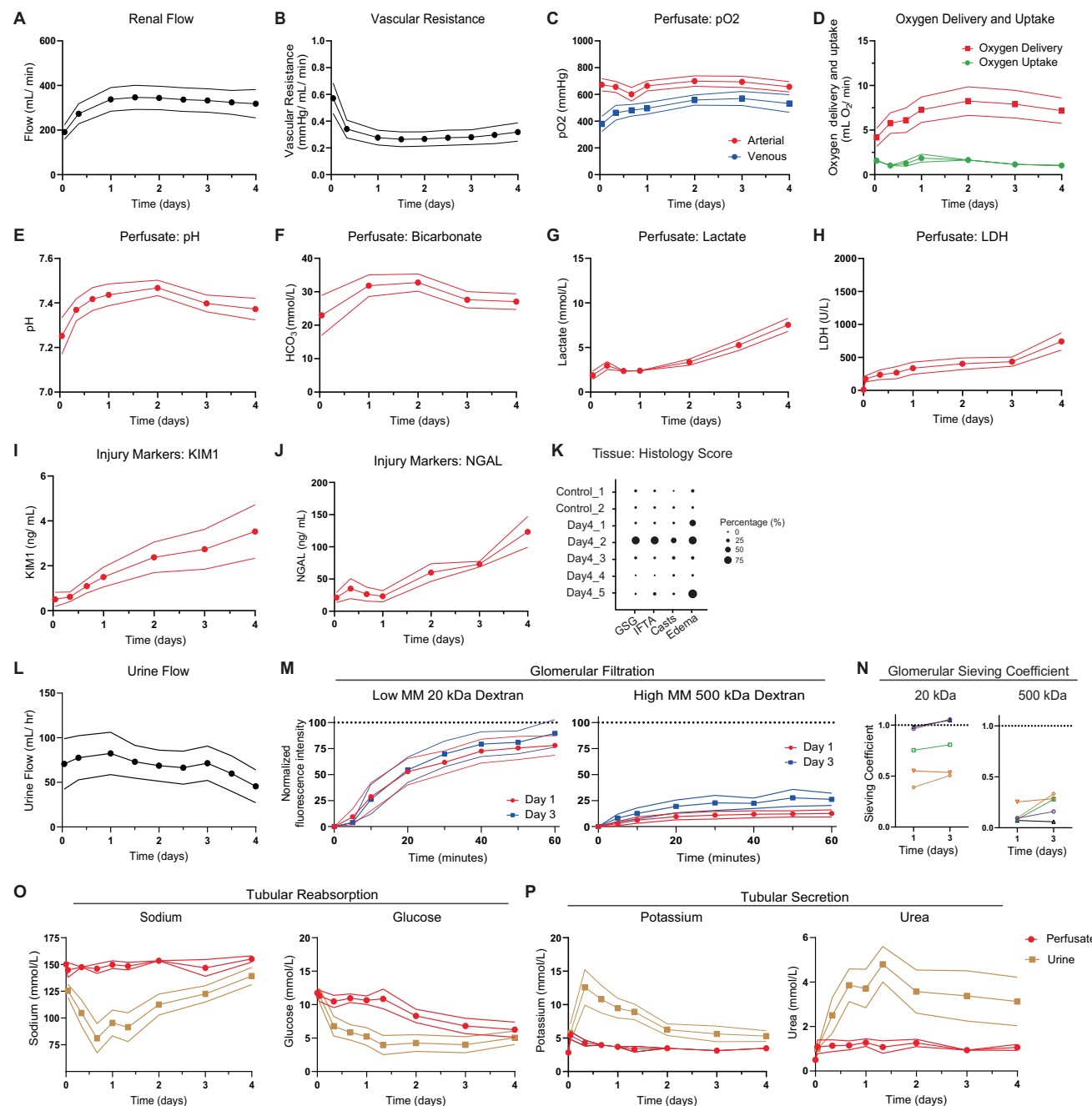

**Fig. 6 | Perfusion dynamics and function during 4-day subnormothermic culture of eight human kidneys.** Course of renal flow (**A**) and vascular resistance (**B**) during the 4-day perfusion period. **C**–**G** Metabolic perfusate dynamics. Perfusate pO₂ ($pO_2$) (**C**) measured in the arterial inflow and venous outflow. Oxygen delivery (**D**) as calculated from the $pO_2$ in the arterial inflow and oxygen uptake (**D**) as calculated from the difference in $pO_2$ between the arterial inflow and venous outflow. Perfusate pH (**E**), perfusate bicarbonate (**F**), and perfusate lactate (**G**) as measured in the arterial inflow. **H**–**J**, Perfusate injury markers during the 4-day perfusion period. Perfusate LDH (**H**) as marker for general cell damage. Perfusate KIM1 (**I**) as marker for proximal tubular cell damage. Perfusate NGAL (**J**) as marker for distal tubular cell damage. **K** Tissue histology scoring of the five discarded human kidneys that were cultured for a 4-day period. Day4_1 and Day4_2 are the contralateral kidneys

of Control_1 and Control_2, respectively. Representative PAS staining's can be found in Figure S4. **L**–**P**, Renal function during 4-day perfusion. **L** Urine flow. **M** Assessment of glomerular filtration barrier on Day 1 and Day 3 of perfusion. Low molecular mass 20 kDa fluorescent dextran (FITC-labeled) and high molecular mass 500 kDa fluorescent dextran (TRITC-labeled) were infused with subsequent perfusate and urine collection (*n* = 5 human kidneys). **N** Glomerular sieving coefficient for 20 kDa and 500 kDa dextran (*n* = 5 human kidneys). For **M**, **N** fluorescence intensity of dextran in the urine was normalized for fluorescence intensity of dextran in the perfusate. Perfusate and urine concentration differences demonstrates tubular reabsorption of sodium and glucose (**O**) and secretion of potassium and urea (**P**) during perfusion at subnormothermia (25 °C). Data are presented as mean ± SEM. Source data are provided as a Source Data file.

## Discussion

In this study, we developed a kidney culture platform that integrates cell-free nutrient-supplemented perfusion at subnormothermia (25 °C) with urine recirculation, and continuous hemofiltration. Eight discarded human kidneys were perfused up to 8 days. Using spatially

resolved single cell resolution isotope tracing we demonstrate active metabolism in the different renal cell types over this period. However, beyond Day 4-Day 6 of perfusion, we observed significant changes in lipid composition in nephron segments, as assessed through spatial lipidomics. In parallel, considerable increases in perfusate injury

markers and oxidative stress were observed beyond Day 4. Up to 4-days perfused human kidneys maintained functional parameters.

Characterization of metabolic and lipidomic changes during multi-day perfusion was performed through MALDI-MSI, allowing the simultaneous mapping of hundreds of lipids and metabolites per pixel while providing spatial information at subcellular resolution ($5 \times 5 \, \mu m^2$ pixel size). Combining this with isotope tracing made it possible to observe cell-type-specific metabolic changes during multi-day perfusion[24]. Metabolic preservation was assessed based upon $^{13}C_6$-Glucose and $^{13}C_5$-Glutamine nutrient partitioning. This yielded unexpected insights into the metabolic activity at different timepoints following prolonged ex vivo preservation. Contrary to an expected decrease in %-enrichment of $^{13}C$-labeled downstream metabolites on Day 8 of perfusion as compared to Day 0 and Day 4, we observed persistent metabolic activity in both glycolysis and TCA cycle (Fig. 5E, F). This finding was surprising considering the accumulation of cell injury (Fig. 1H−K), decrease in oxygen uptake between Day 4 and Day 8 (Fig. 1D), increased lactate abundance (Fig. 1F), and decreased perfusate pH (Fig. 1E). However, the initial spatial heterogeneity in TCA metabolism among different tubular epithelial segments was lost after 4 days, and some reductive TCA activity could not be ruled out. To further assess preservation of mitochondrial metabolism, we performed Q-flux analysis, a recently described method[34], which demonstrated no differences in the relative rate of TCA cycle enzymes between Day 0 and Day 4 of perfusion (Fig. 5O).

All kidneys maintained functional parameters up to Day 4 of perfusion. The glomerular filtration barrier was assessed through the infusion of small (10 kDa) and large (500 kDa) molecular weight (MW) labeled Dextrans. Large MW dextrans were retained within the vascular compartment, whereas the small MW dextrans were filtered into the urine (Fig. 6M, N). Active tubular transport machinery throughout the 4-day period was confirmed by electrolyte gradients between perfusate and urine (Fig. 6O, P). Sodium and glucose were reabsorbed, whereas potassium and urea were excreted. Renal reabsorption and secretion was comparable if not more pronounced during subnormothermic perfusion (sNMP) as compared to a recent case report describing 48 h normothermic human kidney perfusion[37]. Our data demonstrating active metabolism and functionality up to 4 days after procurement can also be seen as remarkable from the perspective that these were discarded human kidneys that had pre-existing damage and injury. For example, one of the kidneys was a retransplant organ.

Metabolic activity is crucial for maintaining the intricate balance of electrolytes, pH, and waste product removal necessary for proper kidney function. Preserving the metabolic processes within the kidneys is therefore essential for future organ donor function. In the field of organ transplantation, ex vivo normothermic perfusion has been widely employed by most research groups as a strategy to preserve metabolic activity of the donor organ, and has been described up to 48 h for the kidney using RBC-based perfusates[38]. However, although this approach has proven promising for liver preservation, it cannot be directly applied to the kidney due to the inevitable occurrence of haemolysis and subsequent acute kidney injury when RBCs are used for oxygenation[11,12]. Brasile et al. pioneered cell-free perfusion of kidneys at subnormothermia two decades ago[15,17,39]. Their research findings indicated that subnormothermic kidney perfusion, as opposed to cold preservation, could effectively mitigate reperfusion injury and recover function ex vivo, while avoiding the necessity of RBCs as an oxygen carrier[39–41]. This requires, however, careful supplementation of nutrients to maintain cell metabolism. The kidney's unique anatomy, where energy-demanding mass solute transport in the renal cortex is coupled to a hypoxic medullary environment that facilitates urine concentration through a countercurrent system, gives rise to significant metabolic heterogeneity. To address these complex metabolic changes, we here applied a spatial approach that allows for an integrated and simultaneous assessment of metabolic changes in different cell types at various time points during the organ preservation process. Our approach not only focuses on the epithelial metabolism, but also examines changes in neighboring endothelial cells, resident immune cells, and stromal cells. By using this approach, we could gain a comprehensive understanding of the metabolic changes that occur in different cell types and their interplay during organ preservation and develop a scientific basis for nutrient suppletion during perfusion.

The question whether these kidneys can be transplanted naturally arises, to provide definitive evidence of their suitability for transplantation. Unfortunately, conducting such an experiment is at this stage unfeasible. The human kidneys utilized in our study were formally discarded and are ineligible for transplantation under Eurotransplant regulations. While one could argue that porcine kidney autotransplantation could serve as substitute, our own investigation into this area showed profound differences in perfusion dynamics between porcine and human kidneys (Supplementary Fig. 6). Because of these differences, we are of the opinion that the outcomes of porcine autotransplantation following multi-day sNMP will not accurately predict the outcomes of human kidney transplantation. We believe that our current assessment of metabolic and functional preservation of human kidneys following multi-day sNMP lays the groundwork for further exploration of clinical application in transplantation medicine. One option is to follow the legal and ethical framework that has been developed to assess xenograft kidney donor organs that were transplanted in brain death recipients, to assesses function after 4-day sNMP[42,43]. Alternatively, a step-wise approach may be of interest, starting with relatively short periods and extending perfusion time once safety in small cohorts of recipients has been demonstrated after transplantation. The latter approach is currently being employed to evaluate the safety and feasibility of normothermic kidney perfusion (ISRCTN13292277 and NCT04693325).

In conclusion, our study showcases that using a cell-free nutrient-supplemented perfusate enables extended ex vivo preservation of metabolically active kidneys. By preserving kidneys in a metabolically active state for days rather than hours, we open possibilities for further advancements in transplantation. Existing protocols already direct donor organs through specialized facilities known as Organ Perfusion and Regeneration-units for assessment and potential pre-treatment using short periods of (cold) machine perfusion. Remarkable breakthroughs in liver preservation highlight the vast potential for prolonged graft preservation, such as the successful transplantation of previously discarded livers[44], transplantation after 3-day ex vivo preservation[9], immunomodulation during machine perfusion[10], and even bile duct regeneration through cholangiocyte organoid transplantation[45]. These advances underscore the transformative impact of prolonged organ preservation and demonstrate the game-changing potential in the field of transplantation.

## Methods
### Human kidneys
This research complies with all relevant ethical regulations. Leiden University Medical Center received authorization from the Dutch government (BWBR0008974, 2555663-CZ/IZ/2562427) for kidney transplantation and associated research. Prior to organ retrieval for all human kidneys, donor research consent was obtained by Eurotransplant, the centralized donation organization in The Netherlands (BWBR008066, Art13). This consent was acquired by an independent organ donation coordinator unaffiliated to the research team.

A total of eight human kidneys were obtained for organ culture using the described platform, after being declined for transplantation because of various reasons (Supplementary Table 2 for donor data). After in situ flushing of the abdominal organs with cold University of Wisconsin (Belzer UW) preservation solution, the kidneys were retrieved and transported to the Leiden University Medical Center on SCS or Hypothermic Machine Perfusion (HMP). Upon arrival at the

laboratory, the renal artery, vein and ureter were cannulated using Luer lock connectors (*Cole Parmer, Barendrecht, the Netherlands*) whilst the organ culture platform was setup in parallel. Excessive renal fat was removed if necessary. Shortly before the start of culture kidneys were flushed with approximately 250 mL of cold DMEM F12 to remove the preservation solution and globally disinfected with betadine. Within the culture platform the renal artery, vein and ureter were connected to their designated outlets.

## Perfusion platform and protocol for human kidneys

*Perfusion platform components.* The platform consists of a closed-loop circuit connected to a custom-designed air tight organ chamber (Mascal Design) that also serves as reservoir. A centrifugal pump (Masterflex L/S Digital Drive 600 rpm) perfused the renal artery through silicone tubing (LS25, Masterflex Metrohm) at a mean arterial pressure (MAP) of 75 mmHg. Pressure in the renal artery was measured in-line with a single-use pressure sensor (Edwards Lifesciences). In the design of the platform we specifically opted for disposable components that have been clinically validated for long-term usage. The perfusate was oxygenated with a carbogen mixture of 95% $O_2$ and 5% $CO_2$ through a long-term membrane oxygenator (Maquet Quadrox-I neonatal, Gettinge; or Lilliput 2 ECMO, LivaNova) that was connected to a water bath set at set at 25 °C.

With urine being recirculated a Prismaflex system (Baxter) was connected in parallel to remove metabolic waste products and supply fresh nutrients whilst maintaining electrolyte homeostasis. A pediatric filter (Prismaflex HF20 set, Baxter) allowed the exchange of small molecular weight molecules. Blood flow through the filter was set at 20 mL/min. Fluid and small molecular weight molecules were removed at 40 mL/h over the filter. Fresh perfusate was substituted post-filter at the same rate. Thus continuous hemofiltration was performed at a filtration fraction of 5%.

Two in-line blood-gas sensor (CDI 500 system, Terumo Cardiovascular Systems) allowed continuous monitoring of $pO_2$, $pCO_2$, pH and temperature in the arterial inflow and venous outflow. Perfusate sampling ports were connected through a 3-way tap (Discofix, Braun) present on the arterial-, venous, and ureteral in-and outlet.

*Perfusate components.* A detailed description of the perfusate composition is provided in Supplementary Table 1. Organ culture was commenced with a total perfusate volume of approximately 700 mL. In brief, DMEM/F-12 (Gibco) was supplemented with Human Serum Albumin (Alburex 20, CSL Behring bv), Insulin-Transferrin-Sodium Selenite (ITS 100x, Sigma-Aldrich), sodium bicarbonate (Gibco), citric acid (Calbiochem, Merck), acetic acid (EMSURE, Merck), penicillin-streptomycin (Gibco), ciprofloxacin (Fresenius Kabi) and fungizone (Bristol-Myers Squibb). pH was adjusted with sodium hydroxide (Calbiochem, Merck) until within range (Target 7.30–7.45). Sodium levels were adjusted with distilled water (Sterile water, Versylene Fresenius) until within range (Target: 130–145 mmol $L^{-1}$).

Continuous hemofiltration was performed at 40 mL/h. Substitution solution was added post-hemofilter by the Prismaflex system. The substitution solution contained the same components as the culture perfusate except for Human Serum Albumin (for composition see Supplementary Table 1). Additional glucose (1 M solution; D-Glucose, G8270, Sigma) was supplemented when perfusate glucose levels dropped below 4 mmol $L^{-1}$, with the goal of maintaining euglycemia.

*Perfusate sampling and measurement.* Perfusate sampling was performed at least three times per day throughout the culture period. Arterial, venous and urine samples were measured directly for Blood-gas analyses and monitoring of electrolytes and metabolites. Samples were aliquoted into Falcon tubes (1 mL) and frozen at −20 °C and/or −80 °C for later analysis.

An iSTAT1 blood analyzer (Abbott) was used for point-of-care blood-gas analysis (pH, $pO_2$, $pCO_2$, $HCO_3^-$). Perfusate samples from the arterial inflow and venous outflow were immediately measured using

CG4$^+$ iSTAT cartridges (Abbott). Blood-gas analysis data were used to calibrate the in-line shunt sensors (CDI-500 system, Terumo Cardiovascular Systems). Oxygen delivery (mL $O_2$ min$^{-1}$) was calculated as Arterial $pO_2$ (mmHg) x Solubility of $O_2$ (0.0031 mL $O_2$ dL fluid$^{-1}$ mmHg$^{-1}$) x Renal Flow (dL min$^{-1}$). Oxygen uptake (mL $O_2$ min$^{-1}$) was calculated as $\Delta pO_2$ (mmHg) x Solubility of $O_2$ (0.0031 mL $O_2$ dL fluid$^{-1}$ mmHg$^{-1}$) x Renal Flow (dL min$^{-1}$). CG4 + -measured $pO_2$ values were used for calculation of oxygen delivery and uptake. Perfusate and urine electrolyte (Sodium, Potassium, Chloride) and metabolite (Glucose, Lactate, Urea) concentration were measured using Chem8 iSTAT cartridges (Abbott). The Clinical Chemistry Laboratory within the LUMC (re)measured sodium, potassium, urea and glucose within arterial perfusate and urine samples according to standard operating procedures, that had previously been stored at −80°C. LDH concentration was measured in perfusate samples.

Neutrophil gelatinase-associated lipocalin (NGAL) and kidney injury molecule-1 (KIM1) levels in the perfusate were measured using a quantitative sandwich enzyme immunoassay technique with NGAL Quantikine ELISA kit (DLCN20, R&D systems) and KIM-1 DuoSet ELISA kit (DY1750B, R&D systems) according to manufacturer's instruction in samples that had previously been frozen and stored at −80 °C. A thiobarbituric acid reactive substances (TBARS) assay kit (Cayman-Chemical, 10009055) was used to measure levels of MDA in the perfusate according to manufacturer's instructions in samples that had previously been frozen and stored at −80 °C.

*Glomerular sieving coefficient.* Filtration of high (500 kDa) and low (20 kDa) molecular mass dextran molecules was assessed to evaluate the intactness of the glomerular basement membrane during ex vivo preservation in the five human kidneys that were cultured for four days[46]. 500 kDa TRITC labeled dextran (0.1 mg/ml) (Sigma-Aldrich 52194) and 20 kDA FITC labeled dextran (0.01 mg/ml) (Sigma-Aldrich 95648) were added to the perfusate after 24- and 72 h of culture. Arterial and urine perfusate samples were taken every 10 min after dextran infusion for 1 h. The fluorescence signal of the arterial and urine perfusate samples was directly measured using a Spectramax M5 (Molecular devices) at wavelengths 495-525 and 555-575.

## Tissue sampling and processing

Tissue punch biopsies (4 mm) were taken at different time points during 8-day organ culture (Day 0, Day 2, Day 4, Day 6, Day 8). At each timepoint biopsies were cut longitudinally into two pieces of which one was fixed in 4% paraformaldehyde and the other snap frozen in liquid nitrogen and stored at −80 °C for further analysis. On Day 0, Day 4 and Day 8 additional biopsies were taken for dynamic metabolic measurements. Tissue biopsies were placed into culture plates and incubated in a well-defined medium (Glucose free and glutamine free DMEM medium (Gibco, A1443001)), supplemented with 2% FCS, 5 mM glucose, 500 μM glutamine and penicillin/streptomycin (pH adjusted to 7.4) for 2 h at 37 °C and 5% $CO_2$. For the $^{13}$C-labeling incubation, same amounts of either U-$^{13}C_6$-glucose (99%, Sigma, 389374) or U-$^{13}C_5$-glutamine (99%, Cambridge Isotope Laboratories, Inc. CLM-1822-H) were used to replace similar un-labeled nutrients in each medium. In the end, tissue slices were quenched with liquid $N_2$ and stored at −80 °C for further analysis.

No biopsies were taken before or during the perfusion of the five human kidneys that were cultured for a 4-day period. From all eight human kidneys cultured within our platform wedge biopsies were taken at the end of culture and fixed in 4% paraformaldehyde or snap frozen in liquid nitrogen and stored at −80 °C. For two of the five human kidneys that were cultured for a 4-day period we received the contralateral kidney from which control wedge biopsies were taken (Control_1 and Day4_1, Control_2 and Day4_2 in Fig. 6K and Fig. S4A).

## Histology

Biopsies were fixed overnight in 4% paraformaldehyde, stored in 70% ethanol and embedded in paraffin for subsequent sectioning. Periodic

Acid-Schiff (PAS) staining was performed on 4-μm-thick paraffin embedded cortical tissue sections by the Pathology department using an automatic slide stainer. Tissue slices that are compared were processed simultaneously. Bright field images were digitized using a 3D Histech Pannoramic MIDI Scanner (Sysmex) and viewed with Case-Viewer software. PAS stained sections of the five four-day cultured kidneys were assessed by a renal pathologist and scored on globally sclerosed glomeruli (GSG), Interstitial Fibrosis and Tubular Atrophy (IFTA), tubular cast formation, and edema.

## MALDI-MSI

Tissue preparation, matrix deposition, MALDI-MSI measurements, post MALDI-MSI staining, and data processing and analysis were performed as previously described[23,24]. Details are provided below. In total, 33 sections were assessed through MALDI-MSI, comprising a total of 2.394.033 pixels analyzed for this study, resulting in an average of 72.546 pixels per section at a spatial resolution of $5 \times 5\,\mu m^2$. The average measured area for one section was $1.8\,mm^2$.

### Tissue preparation and matrix deposition

Cryo preserved tissue biopsies were embedded in 10% gelatin and cryosectioned into 10-μm-thick sections using a Cryostar NX70 cryostat (Thermo Fisher Scientific) at $-20\,°C$. The sections were thaw-mounted onto indium-tin-oxide (ITO)-coated glass slides (VisionTek Systems) and stored at $-80\,°C$ until further use. Slides were placed in a vacuum freeze-dryer for 15 min prior to matrix application. After drying, $N$-(1-naphthyl) ethylenediamine dihydrochloride (NEDC) (Sigma-Aldrich, UK) MALDI-matrix solution of 7 mg/mL in methanol/acetonitrile/deionized water (70/25/5% v/v/v) was applied using a SunCollect sprayer (SunChrom GmbH). A total of 21 matrix layers was applied at the following flow rates: layers 1–3 at 5 μL/min, layers 4–6 at 10 μL/min, layers 7–9 at 15 μL/min and layers 10–21 at 20 μL/min (speed x, medium 1; speed y, medium 1; z position, 35 mm; line distance, 1 mm; $N_2$ gas pressure, 35 psi).

### MALDI-MSI measurement

MALDI-TOF/TOF-MSI was performed using a RapifleX MALDI-TOF/TOF system (Bruker Daltonics). Negative-ion-mode mass spectra were acquired at a pixel size of $5 \times 5\,\mu m^2$ over a mass range of $m/z$ 80–1000. Prior to analysis, the instrument was externally calibrated using red phosphorus. Spectra were acquired with 15 laser shots per pixel (for $5 \times 5\,\mu m^2$ measurements) at a laser repetition rate of 10 kHz. Data acquisition was performed using flexControl (Version 4.0, Bruker Daltonics) and flexImaging 5.0 (Bruker Daltonics). Sections present on the same slide were measured in a randomized order. The $m/z$ features present in MALDI-TOF-MSI dataset were further used for identity assignment of metabolites and lipid species. The $m/z$ values were imported into the Human Metabolome Database[47] (https://hmdb.ca/) after re-calibration in mMass and annotated for metabolites and lipids species with an error $\leq \pm 20$ ppm. The $^{13}C$-labeled peaks were selected by comparing the spectrum of control and $^{13}C$-labeling experiments, and annotated based on the presence of un-labeled metabolites and their theoretical $m/z$ values. Peak intensities of the selected features were exported for all the measured pixels from SCiLS Lab 2016b (version 2016b, Bruker Daltonics), which were used for the following analysis. Single ion visualizations were also obtained from SCiLS Lab.

### Post-MALDI-MSI staining

Following the MALDI-MSI data acquisition, excess matrix was removed by washing the slides in 100% ethanol (2 × 5 min), 75% ethanol (1 × 5 min), and 50% ethanol (1 × 5 min), after which tissues on the slide were fixed using 4% paraformaldehyde for 10 min. For immuno-fluorescent staining, slides were blocked with 3% normal donkey serum, 2% BSA, and 0.01% Triton X-100 in PBS for 1 h at room temperature. Primary lotus tetragonolobus lectin (LTL, 1:300, Vector

laboratories, B1325), anti-nephrin (5 μg/mL, R&D, AF4269), and anti-CDH1 antibody (1:300, BD Biosciences, 610181) were incubated overnight at 4 °C, followed by correspondent fluorescent-labeled secondary antibodies for 1 h at room temperature Streptavidin anti-biotin AF488 (1:300, ThermoFisher S-11223), Donkey anti-sheep IgG AF568 (1:300, Invitrogen, A-21099), Goat anti-Mouse IgG2a AF647 (1:300, ThermoFisher, A-21241). Slides were embedded in Prolong gold anti-fade mountant with DAPI (Thermo Fisher Scientific, P36931). The stained tissues were scanned using a digital slide scanner (3D Histech Pannoramic MIDI Scanner, Sysmex). Digital scanned images were aligned with the MALDI-MSI data.

### MSI data processing and analysis

For lipid analysis, features with $m/z \geq 400$, predominately glycerophospholipids, that did not co-localize with MALDI matrix signals were selected (signal-to-noise-ratio $\geq 3$). The per-pixel total ion count (TIC)-normalized intensity values for each $m/z$ feature from all MSI measurements were directly exported as comma-separated values (.csv format). Upon loading in R (v. 4.0), these values were transformed into a count matrix for UMAP analysis by multiplying the intensities by 100 and taking the integer. This count data matrix was normalized and scaled using SCTransform to generate a 2-dimensional UMAP projection using Seurat[48]. The spatial reconstructions of the segmentation clusters were compared to the aligned immunofluorescence stainings, and cell types were identified based on both immunomarker staining and tissue morphology. Same cell types were annotated within one cluster. The differential abundance of lipids between clusters were analyzed using the FindAllMarkers function in Seurat. To compare pixels from different datasets, these matrices were imported into the Seurat package and a data integration step was performed after batch correction using the method provided by Seurat. These integrated datasets were used to generate a 2-dimensional UMAP projection using Seurat and 3-dimensional UMAP projection using the Seurat and plotly packages. The embedding information of the 3-dimensional UMAP was translated to RGB color coding by varying red, green and blue intensities on the 3 independent axes. Together with pixel coordinate information exported from SCiLS Lab, a MxNx3 matrix was generated and used to generate molecular histology images in Matlab (v. R2019a.; Mathworks).

For dynamic metabolomics, the lipid $m/z$ features from the control kidneys were used as query. Then the MALDI-MSI data from $^{13}C$-labeling experiments of Day_0, Day_4 and Day_8 human kidneys were used as a reference to transfer metabolite production into the query using FindTransferAnchors and TransferData function from Seurat package. Both the query and reference were normalized and scaled using SCTransform. Ultimately, all the imputed metabolite productions were combined into control kidney dataset, which contained the $^{13}C$-labeling information from different nutrients. The $^{13}C$-labeled metabolite abundance was corrected to its isotope tracer purity. Natural isotope abundance correction was performed for metabolites using R package IsoCorrectoR[49]. After combining all the imputed data into one dataset and correcting the natural isotope abundance, the fraction enrichment of isotopologues was calculated based on the ratio of each $^{13}C$-labeled metabolite (isotopologue) to the sum of this metabolite abundance in each pixel. The calculated fraction enrichment of isotopologues was used to generate pseudo-images together with pixel coordinate information exported from SCiLS Lab. The average fraction enrichment values of identified clusters were used for generating graphs and statistical analysis. Hotspot removal (high quantile 99%) were applied to all the pseudo-images generated from calculated values.

The fraction enrichment of isotopologues derived from $^{13}C_5$-glutamine were further used for relative flux rate calculation according to the previous published Q-Flux equations[34]. Previously we demonstrated that after 2 h of ex vivo incubation with $^{13}C_5$-glutamine a

pseudo-steady state is reached, allowing Q-flux analysis[24]. By comparing the glutamate $(m+5)$ enrichment and glutamine $(m+5)$ enrichment, the rate of GLS relative to OGDH ($V_{GLS}/V_{OGDH}$) was determined by Eq. 1. Next, comparing the two fractional contributions of α-KG derived from glutamine and IDH flux yields the rate of GLS relative to IDH ($V_{GLS}/V_{IDH}$) (Eq. 2). The fraction of succinate derived from α-KG /glu defines $V_{OGDH}/V_{SDH(F)}$ (Eq. 3). The fraction of malate derived directly from SDH forward flux was determined by dividing malate $(m+4)$ enrichment by succinate $(m+4)$ enrichment (Eq. 4).

$$\frac{V_{GLS}}{V_{OGDH}} = \left( \frac{\text{glutamate} \, (m+5)}{\text{glutamine} \, (m+5)} \right) \tag{1}$$

$$\frac{V_{GLS}}{V_{IDH}} = \left( \frac{\text{glutamate} \, (m+5)}{\text{glutamine} \, (m+5)} \right) \Big/ \left( 1 - \left( \frac{\text{glutamate} \, (m+5)}{\text{glutamine} \, (m+5)} \right) \right) \tag{2}$$

$$\frac{V_{OGDH}}{V_{SDH(F)}} = \left( \frac{\text{succinate} \, (m+4)}{\text{glutamate} \, (m+5)} \right) \tag{3}$$

$$\frac{V_{SDH(F)}}{V_{MDH}} = \left( \frac{\text{malate} \, (m+5)}{\text{succinate} \, (m+5)} \right) \tag{4}$$

## Untargeted lipidomics

*Sample preparation for kidney lipid extraction.* Frozen kidney tissues were weighted and homogenized in cold LC-MS-grade water (final concentration: 10 mg/mL) with a Bullet Blender™ 24 (Next Advance Inc., NY, USA). Lipid extraction took place by adding 600 μL methyl-tert-butyl ether and 150 μL of methanol to 25 μL of tissue homogenate. Samples were vortexed and kept at room temperature for 30 min. After centrifugation, the supernatant was transferred to a new microtube. The extraction was repeated with 300 μL methyl-tert-butyl ether and 100 μL methanol after which the supernatants were combined and 300 μL of water was added before centrifugating once again. The upper (nonpolar) phase was transferred to glass vials and was dried under a gentle stream of $N_2$. Samples were reconstituted in 100 μL 2-propanol, sonicated for 5 min and 100 μL water was added and sonicated once again for 5 min. Finally, samples were transferred to micro-vial inserts and placed in the autosampler for LC-MS/MS analysis. Quality controls (QC) were made by combining 15 μL from each sample.

*Untargeted lipidomic analysis.* Kidney lipid extracts were analysed using a LC-MS/MS based untargeted lipid profiling method[50]. The LC system was a Shimadzu Nexera X2 (consisting of two LC30AD pumps, a SIL30AC autosampler, a CTO20AC column oven and a CBM20A controller) (Shimadzu, 's Hertogenbosch, The Netherlands). The mobile phases consisted of water:acetonitrile 80:20 (eluent A) and water:2-propanol:acetonitrile 1:90:9 (eluent B), both eluents containing 5 mM ammonium formate and 0.05% formic acid. The following gradient was applied using a flow rate of 300 μL/min: 0 min 40% B, 10 min 100% B, 12 min 100% B. A Phenomenex Kinetex C18, 2.7 μm particles, 50 × 2.1 mm (Phenomenex, Utrecht, The Netherlands) column with a Phenomenex SecurityGuard Ultra C8, 2.7 μm, 5 × 2.1 mm cartridge (Phenomenex, Utrecht, The Netherlands) as guard column were used. The column was kept at 50 ˚C at all times and the injection volume was 10 μL. The MS system was a Sciex TripleTOF 6600 (AB Sciex Netherlands B.V., Nieuwerkerk aan den IJssel, The Netherlands) operated in negative ESI mode (ESI-) using the following parameters: ion source gas 1 45 psi, ion source gas 2 50 psi, curtain gas 35 psi, temperature 350 ˚C, acquisition range $m/z$ 100-1800, ion spray Voltage −4500 V (ESI-), declustering potential −80 V (ESI-). Information dependent acquisition (IDA) was used to identify lipids, with the following conditions for MS analysis: collision energy −10 V and acquisition

time 250 ms; for MS/MS analysis: collision energy −45 V, collision energy spread 25, ion release delay 30, ion release width 14 and acquisition time 40 ms. The IDA switching criteria were set for ions greater than $m/z$ 300 that exceed 200 cps, excluding isotopes within 1.5 Da resulting in maximum 20 candidate ions, including a selection of oxidized phospholipids (oxPLs) as inclusion list. MS-DIAL (v5.1) was used to align the data and identify the different lipids[51–53]. In brief, all peaks with an intensity of at least 200 ions eluting between 0.5 and 10 min of the chromatogram were considered. Alignment within samples was performed with a retention time and MS1 tolerance of 0.15 min and 0.025 Da, respectively. MS-DIAL lipid database version Msp20221205132019 was used for lipid annotation having the following parameters as identification settings: 0.01 and 0.05 Da for both MS1 and MS2 accurate mass tolerance, respectively, and including [M-H]-, [M-H2O-H]- and [M+formic acid-H]- as adducts. Manual curation for the oxPLs species was performed by only including those oxPLs for which the experimental and reference MS/MS spectra matched. The annotation of any other lipid class has not yet been verified as this was out of the scope of this work. Subsequently, matching of untargeted lipidomics data with the results obtained from MALDI-TOF and MALDI-FTICR MSI analysis was based on matching accurate masses (5 ppm) of consistently observed oxPL.

## Kidney auto-transplantation

Experiments were performed at the Toronto Organ Preservation Lab (TOPL), following a previously established protocol for porcine kidney procurement and auto-transplantation[35,36,54,55]. These experiments were carried out in accordance with the Canadian Council on Animal Care guidelines. Animal ethical approval was granted under an Animal User Protocol (AUP) issued by the University Healthcare Network Animal Care Committee (AUP number 3651).

Male Yorkshire pigs (30−32 kg, 3-months-old) were used. Operative and perioperative procedures, drug administration, and follow-up were conducted as described previously[35,55]. A detailed description is provided in the supplementary materials and methods.

## Statistical analyses

Perfusion parameters are presented as mean ± SEM, unless indicated otherwise. MALDI-MSI derived data are presented as mean ± SD, unless indicated otherwise. Data normality and equal variances were tested using the Shapiro−Wilk test. All statistical tests were performed using GraphPad Prism 9. $P < 0.05$ were considered statistically significant. Exact sample size ($n$) for each experimental group and units of measurements are provided in the text and figure captions. Figures were created using Adobe Illustrator (Adobe Systems).

## Reporting summary

Further information on research design is available in the Nature Portfolio Reporting Summary linked to this article.

## Data availability

All the exported and processed MALDI-MSI data underlying the main text and supplementary materials were deposited in Figshare at https://doi.org/10.6084/m9.figshare.25304326. Owing to the large size of the raw MALDI-MSI data this could not be deposited in a public repository. The full MALDI imaging data generated in this study are available upon request from the corresponding author (please contact Gangqi Wang and Ton Rabelink). Source data are provided with this paper.

## Code availability

The code used for this article has been published previously[24] and is available in Zenodo at https://doi.org/10.5281/zenodo.7191331.

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

## Acknowledgements

We gratefully acknowledge the support and assistance provided by Udo Weiland (LUMC, Leiden, the Netherlands) for arranging the dialysis machines used during perfusion experiments, and Manon Zuurmond (LUMC, Leiden, the Netherlands) for making the illustrations. We would like to thank Sujani Ganesh (Ajmera Transplant Center, Toronto General Hospital, Toronto, Canada) and Lisa Robinson (Division of Nephrology, The Hospital for Sick Children, Toronto, Canada) for their expertize and assistance during the auto-transplantation experiments. Lastly, we express our gratitude to the donors and their families for entrusting us with the use of donor kidneys deemed unsuitable for transplantation, without which this research would not be possible. The work of the authors is supported by the Dutch Kidney Foundation through the Participants of the Friends Lottery (20INI011). The Novo Nordisk Foundation Center for Stem Cell Medicine (reNEW) is supported by Novo Nordisk Foundation grants (NNF21CC0073729).

## Author contributions

Conceptualization: M.dH., M.J., F.W., G.W., M.E., T.R. Data curation: M.dH., M.J., F.W., G.W., E.S.L., S.K., M.G.,. B.H., G.W. Formal analysis: M.dH., M.J., F.W., E.S.L., S.K., M.G., J.K., B.H., G.W. Funding acquisition: T.R.; Investigation: M.dH., M.J., F.W., A.dG., E.S.L., S.K., F.C.N., T.C., M.M., D.dV., B.H., G.W., M.E. Methodology: M.dH., M.J., F.W., S.K., M.G., D.dV., B.H., G.W., M.E., T.R. Project administration: M.dH., G.W. Resources: S.K., M.G., M.M., T.C., F.C.N., M.S., B.H. Software: S.K., M.G., B.H., G.W. Supervision: G.W., M.E., T.R. Visualization: M.dH., G.W. Writing – original draft: M.dH., G.W., T.R. Writing – review and editing: M.dH., M.J., F.W., S.K., M.G., M.M., F.C.N., M.S., D.dV., J.K., I.A., C.vK., B.H., G.W., M.E., T.R.

## Competing interests

The authors declare no competing interests.

## Additional information

¹Department of Internal Medicine (Nephrology) & Einthoven Laboratory of Vascular and Regenerative Medicine, Leiden University Medical Center, Leiden, The Netherlands. ²The Novo Nordisk Foundation Center for Stem Cell Medicine (reNEW), Leiden University Medical Center, Leiden, The Netherlands. ³Center for Proteomics and Metabolomics, Leiden University Medical Center, Leiden, The Netherlands. ⁴Ajmera Transplant Centre, Department of Surgery, University Health Network, Toronto, ON, Canada. ⁵University of Lille, Institut National de la Santé et de la Recherche Médicale (INSERM), Centre Hospitalier Universitaire de Lille (CHU Lille), Institute Pasteur Lille, Lille, France. ⁶Transplant Center, Leiden University Medical Center, Leiden, The Netherlands. ⁷Department of Surgery, Leiden University Medical Center, Leiden, The Netherlands. ⁸Department of Pathology, Amsterdam UMC, University of Amsterdam, Amsterdam, The Netherlands. ⁹Department of Pathology, Leiden University Medical Center, Leiden, The Netherlands. ¹⁰These authors contributed equally: Marlon J. A. de Haan, Marleen E. Jacobs, Franca M. R. Witjas. ¹¹These authors jointly supervised this work: Gangqi Wang, Marten A. Engelse, Ton J. Rabelink. ✉e-mail: g.wang@lumc.nl; m.a.engelse@lumc.nl; a.j.rabelink@lumc.nl

