## [Peer Review File · Nature Communications]

A cell-free nutrient-supplemented perfusate allows four-day
ex vivo metabolic preservation of human kidneysEditorial Note: Parts of this Peer Review File have been redacted as indicated to remove third-party material where no permission to publish could be obtained.

REVIEWER COMMENTS

Reviewer #1 (Remarks to the Author):

This study reports on the use of subnormothermic machine perfusion technology to preserve metabolism in human and porcine kidneys for up to 4 days using an acellular perfusate supplemented with TCA cycle intermediates at 25°C. Spatially resolved isotope tracing was used to follow metabolic fluxes.

This manuscript is well written and clearly presented. The topic is extremely novel and important for the future development of perfusion technologies. Kidneys were successfully perfused for up to 4 days. Longer periods of perfused showed that lipid remodelling occurred, there was an increase in oxidative damage, tubular injury, and some loss of the TCA cycle between 6 and 8 days. The authors use lipidomics at a single cell resolution to examine lipidomic and metabolic changes.

1. Where any of the components of the perfusion circuit changed during perfusion? How many days is the membrane oxygenator and dialysis filter able to work efficiently for? Although oxygen delivery and uptake were stable, actual tissue oxygen tension is not measured. Furthermore, oxygen consumption decreased and oxidative damage increased. Would an oxygen carrier in the perfusate not be necessary for oxygen delivery for more prolonged periods of perfusion?
2. It would be important to demonstrate that kidneys can be transplanted successfully after 4 days of ex vivo perfusion. The porcine model fits this well but the authors suggest that in their own experience this yielded limited information. Is this because the animals did not recover kidney function? A 3 hour period of reperfusion after transplanted is a very limited time to make any assessment of kidney function. A longer recovery phase would be important to demonstrate viable kidney function.
3. Perfusion parameters were stable, but it appears that even over the first 4 days there is

evidence of the accumulation of damage. Increasing levels of lactate, KIM-1 and NGAL. The scale bars in figure 6 are deceiving for values of KIM-1 and NGAL. The levels of KIM-1 rise to approximately 4ng/mg which is considered high.

4. What was the level of cold ischaemic in the discard kidneys. Two of the kidneys used for the 8 day ex vivo perfusion have significant kidney dysfunction at the time of donation. Can the authors comment? Was this injury apparent during perfusion?

Minor comments

1. Kidneys gained weight but with no evidence of oedema. The authors speculate that this indicated the vascular barrier function was preserved. Simply weighing the kidney is not an accurate measure. Wet/dry ratio would be more exact.
2. The group in Oxford have shown stable perfusion with red cells for up to 48h. This should be included in the discussion.

.

Reviewer #2 (Remarks to the Author):

I read with interest the manuscript: NCOMMS-23-12780A-Z intitled: A cell-free nutrient-supplemented perfusate allows four-day ex vivo metabolic preservation of human kidneys.

This work is of obvious interest, with a first-rate technological approach. The aim is to infuse human discarded kidneys over several days using a home-made infusion circuit. In this study, the perfused human kidneys discarded for transplantation with a cell-free perfusate supplemented with TCA cycle intermediates at sub normothermia (25°C) up to 8 days.

The primary objective of this study was the development of a perfusion loop that supports multi-day ex vivo preservation of metabolically active human kidneys. The authors have followed a very demanding set of specifications for a model capable of meeting the constraints of organ perfusion and fulfilling their ambition.

The methodology uses renal tissue recovered during perfusion for metabolic and functional

analysis. Krebs cycle metabolites were added to study metabolic activity. The perfusion medium is described, with particular reference to the metabolites involved (glutamine, citrate and acetate). This prototype perfusion circuit Kidney perfusion and the parameters characterizing it are presented with their evolution during the 8 days and highlight modifications from 4 days onwards in terms of performance. The monitoring of lesion markers also shows a progressive evolution with a potential correlation with histological studies. I am concerned by 30% kidney weight gain (Day0: 283±39, Day8: 363±52 gram) which is intriguing and difficult to explain despite comments minimizing this weight gain.

As far as donors are concerned, there is considerable variability in terms of cause of death and different clinical characteristics, which must be taken into account. LDH and creatinine levels are heterogeneous. In this context, the GFR estimation may not be relevant taking into account the period.

Assessments by the various techniques (MALDI-MSI measurement, Post-MALDI-MSI staining, Untargeted lipidomics) provide highly innovative and functional information of great importance.

Mass Spectrometry Imaging (MSI) is a surface mapping technology that allows the spatial information and relative abundance of analytes (proteins, peptides, lipids or small metabolites) to be determined directly from the surface of a biological sample section. Following MALDI-MSI measurements, we performed Uniform Manifold Approximation and Projection (UMAP) analysis based on the lipidomic data, combined with post-MALDI-MSI immunofluorescence staining for cell type identification. Using this method, the authors were able to distinguish different cell types and their evolution during perfusion during perfusion, with stability between D4 and D6. Unsupervised hierarchical clustering of the different timepoints for each cell population revealed that lipid remodelling predominantly occurred between Day6 and Day8 of perfusion. From Day6 of perfusion onwards, the authors observed a progressive increase in the perfusate levels of malondialdehyde (MDA), an indicator of oxidative stress. The results highlight the disruption of the pro-oxidant balance, with the depletion of anti-oxidant defences. The spatial distribution of these oxidized phospholipids was assessed within the MALDI-MSI analysed biopsies and results

supports that oxidized lipid species are absent between Day0 and Day4-Day6 of ex vivo perfusion, with progressive oxidative stress taking place between Day4-Day6 and Day8 of perfusion resulting in the accumulation of peroxidised lipid species towards Day8. These results lead to two comments: 1. this D4-D6 window needs to be clarified and specified in terms of impact, and 2. this oxidative stress needs to be analyzed in terms of its longer-term consequences.

Metabolic analysis show that all renal cell types identified in the biopsies maintained active cell metabolism and could use glucose and glutamine as nutrient sources for glycolysis and the TCA cycle after 4- and 8-days of perfusion. It was suggested an active glycolysis pathway and the contribution of glucose to the TCA cycle until Day8 of perfusion and a higher contribution of glutamine as carbon source for the TCA cycle as compared to glucose. Finally, conclusion from authors is that their finding underpins that sub normothermic machine perfusion allows multi-day ex vivo metabolic preservation of diseased human donor kidneys.

On the subject of renal function assessment, the existence of diuresis is not a rigorous means of assessing renal function.

Tubular functions are nevertheless modest (despite the optimistic comments), and there is significant natriuresis and glucosuria indicating poor functionality.

The results presented in this work provide very interesting data on the metabolic state of the organs perfused by the system described.

On the other hand, the limitations described for the porcine model, and especially the conditions described with a 3h reperfusion, are fully justified. It would probably have been appropriate to propose a follow-up of one to two weeks.

Furthermore, the use of two different preservation solutions for human and porcine kidneys is not clearly justified and would warrant further clarification.

It would have been useful to present the prospects for the human clinic and how this

platform could be a precursor for evaluation and repair/regeneration structures (HUB concept).

Minor comment: is substation the correct word ?

Reviewer #3 (Remarks to the Author):

De Haan et al: A cell-free nutrient-supplemented perfusate allows for 4 day ex vivo metabolic preservation of human kidneys

The authors performed mass spectrometry imaging analyses in the context of human kidney perfusion ex vivo. The authors combine ex vivo culture, analyses of biopsies with physiological measurements and chiefly with an imaging MS-platform. The use of isotopes is an elegant strategy used here, yet the authors do not demonstrate supporting data that their quantification is sound. In total, the identification of metabolites and their quantification raises some concerns.

The main concern is that the findings the authors present are not corroborated by other methods. Presented as is, conclusions cannot be made regarding the nature of metabolic changes, and appear to be made by a very selected observation of only a few cells. Bulk analyses with additional LC separations are necessary to confidently assign identities to the claimed metabolites and their isotopologues. A global overview of quantification of fluxes is missing. Code and raw data are not provided. The authors provide overall cherry-picked metabolic pathways, chiefly focusing on use of glucose, but not other substrates such as fatty acids, or lactate as fuels.

Major concerns:

1. The high resolution MS imaging approach is generally elegant, and 20ppm should be sufficient to resolve those peaks. However, spectral quality and S/N ratios need to be good enough in order to really make the conclusions the authors make. Aspartate has a mass of 132.02968267, and malate has one of 132.006973 – can the authors show that these two metabolites are really distinguishable without LC? Similar, Glutamate has a mass of

146.04533274, glutamine 146.06914219 – are the spectral data clean enough to demonstrate the different metabolic behavior of these two metabolites?

This is particularly concerning when labels are very low, and noise is likely high i.e. less than 1 % as in Fig. 5N, F. Another concern is the measurement of glutathione (Figure 4D): GSH is so challenging to quantify since it undergoes autooxidation, how are the authors sure that this is really this metabolite?

2. Figure 2: How many images were taken into account for the quantification? I thought that it is expected that the tissue heterogeneity is very high even macroscopically, so several images per kidney are likely necessary? How are the authors sure that their “PT clusters” are not a simple result of over-clustering? Based on how many cells have these analyses been done? It seems the amount of cells is rather limited. The overall area imaged needs to be depicted, not just the number of kidneys.

3. All data are quantified as n=1 without SD/SEM and presented with one sample t-test which is not appropriate if three replicates were analyzed. In Fig3: log₂ fold changes need to be added.

4. Information from the untargeted lipidomics and the MS imaging should be integrated better. Lists/excel files of identified metabolites/features and their kinetics are not provided. The untargeted analysis of lipids could be better described (MS DIAL) – but how exactly were lipids identified and annotated is unclear.

5. I cannot review their bioinformatics pipeline, since no information on bioinformatics is provided. Full data and code transparency is necessary to provide a review of the paper.

6. In Fig. 5 scale bars are vastly different between the individual replicates, giving a false impression of the real signals and isotope ratios. Can the authors apply the same scale bar to all images?

7. PT1,2,3 are behaving very similarly metabolically (Fig. 5) – is this expected? Is it then necessary to distinguish them at all? It seems like the authors do not take advantage of their spatial resolution by looking at other cells such as glomeruli – why not? Again, a larger aggregation of data – instead of looking at 1 panel at a time – would be useful.

8. I am not an expert in ex vivo transplant perfusion, but it seems to me that a “viability in cell culture” /metabolic fitness aspects needs more data to confirm that these kidneys are viable and respiratory capacity is conserved. Why did the authors not elect to use the gold standard for metabolic capabilities, i.e. seahorse assay?

9. The authors use QFlux equations to determine fluxes based on Hubbard et al. Cell Metabolism 2023. These, however, are based on tracer steady state infusion of glutamine in mice. This was probably not the case here. The authors should provide more validation that this approach is really applicable.

Line 468: Figure 3K does not exist.

RESPONSE TO REVIEWERS

Reviewer #1 (Remarks to the Author):

This study reports on the use of subnormothermic machine perfusion technology to preserve metabolism in human and porcine kidneys for up to 4 days using an acellular perfusate supplemented with TCA cycle intermediates at 25°C. Spatially resolved isotope tracing was used to follow metabolic fluxes.

This manuscript is well written and clearly presented. The topic is extremely novel and important for the future development of perfusion technologies. Kidneys were successfully perfused for up to 4 days. Longer periods of perfused showed that lipid remodelling occurred, there was an increase in oxidative damage, tubular injury, and some loss of the TCA cycle between 6 and 8 days. The authors use lipidomics at a single cell resolution to examine lipidomic and metabolic changes.

Response: We thank the reviewer for the positive feedback and the recognition of its importance for the development of future kidney perfusion technologies. We are confident that acellular preservation at subnormothermia holds great potential for improving and prolonging kidney preservation.

1. Where any of the components of the perfusion circuit changed during perfusion? How many days is the membrane oxygenator and dialysis filter able to work efficiently for? Although oxygen delivery and uptake were stable, actual tissue oxygen tension is not measured. Furthermore, oxygen consumption decreased and oxidative damaged increased. Would an oxygen carrier in the perfusate not be necessary for oxygen delivery for more prolonged periods of perfusion?

Response: Thank you for pointing out this important aspect. In the design of the platform, we specifically opted for disposable components that have been validated for long-term usage. We have emphasized this in the methods section (line 403-406). Due to varying availability, we have used two different commercially available long-term oxygenators. Eight-day perfusions were performed with long-term Quadrox-iD oxygenators (Macquet, Getinge Group), which have previously been used up to 12-days during ex vivo liver perfusion (1). The four-day perfusions were either performed with the Quadrox-iD oxygenators or Lilliput 2 ECMO (LivaNova), which is clinically validated for use up to 5 days. We did not observe any changes in the membrane oxygenators oxygen dissolving capacity during the perfusions.

The dialysis filter was reloaded after 72-hours, as required by the clinical Prismaflex dialysis machines that were used.

During initial experiments we assessed the oxygen carrying capacity of our acellular perfusate (i.e. in the absence of oxygen carriers), and oxygen uptake of the kidney at varying temperatures (see Figure S1). This demonstrated that, when perfused at 25°C, sufficient oxygen can be delivered without the need for oxygen carriers. This is in line with previous publications regarding the use of acellular perfusates (2-5). A recent study even demonstrated that acellular normothermic perfusion (i.e. 37°C) at supraphysiological oxygen partial pressure seems to be a safe alternative to RBC-based

perfusion (6). They did not observe differences in energy dependent tubular cell function like sodium and glucose reabsorption between acellular and RBC-based NMP, further demonstrating that sufficient oxygen can be delivered using acellular perfusates.

2. It would be important to demonstrate that kidneys can be transplanted successfully after 4 days of ex vivo perfusion. The porcine model fits this well but the authors suggest that in their own experience this yielded limited information. Is this because the animals did not recover kidney function? A 3 hour period of reperfusion after transplanted is a very limited time to make any assessment of kidney function. A longer recovery phase would be important to demonstrate viable kidney function.

Response: We thank the reviewer for their critical appraisal of the porcine kidney transplantation we previously described. To better address whether porcine transplantation would have been a suitable step towards clinical translation of our platform we set up a novel collaboration with the group of dr. Markus Selzner in Toronto, where they have a well-established porcine kidney auto-transplantation model (7-9).

However, what we found is that the perfusion dynamics of 3-month-old male porcine living-donor kidneys differed considerably from our previous observations in human kidneys (Figure S6-7, and below). During porcine kidney perfusions, vascular resistance began to rise between 12-24 hours, leading to a gradual reduction in renal flow. Furthermore, at the conclusion of perfusion, the porcine kidneys had gained 114-308% compared to their initial weight, a notable contrast to our findings in human kidneys (Figure S6, and below).

Figure. Comparison of perfusion dynamics between porcine kidneys and human kidneys. A, Vascular resistance. B, Renal flow. C, Whole organ weight at start and end of perfusion. D, Weight change during perfusion. E, Vascular resistance dynamics in porcine and human kidneys. F, Renal flow dynamics in porcine and human kidneys. For E and F * denotes the termination of porcine kidney perfusion #2. G, Weight change during porcine and human kidney perfusion.

Despite the variations between these porcine kidney perfusions and the previously perfused human kidneys, both Kidney#1 and Kidney#3 underwent auto-transplantation. Kidney#1 was transplanted following a contralateral nephrectomy. Initially, a uniformly perfused pink kidney was observed after reperfusion. However, in the following hour the kidney developed haemorrhagic infarction (see below) and the pig entered a distributive shock that required large doses of pressors and fluids. We believe the intraparenchymal blood was a reflection of disturbed vascular permeability, as was also supported by the extraordinary weight gain of the kidney during perfusion preservation.

For the transplantation of Kidney#3 the contralateral kidney remained in situ. After reperfusion, a patchy appearance with pink and darker regions was noted, as is often observed in DCD donations. Reperfusion differed from Kidney#1, as minimal pressors and fluids were required to maintain blood pressure. The pig was sacrificed after 7 days. Again, the kidney showed haemorrhagic infarction and intraparenchymal blood, reflective of disrupted vascular permeability, albeit to a lesser extent when compared to kidney#1.

Figure. Macroscopic appearance of the porcine kidneys at different timepoints.

It should be emphasized that such loss of capillary permeability was not noted during the human kidney perfusions. For example, the sieving coefficient for dextrans was maintained and weight gain was modest. Therefore, given these distinct differences in perfusion dynamics, we do not believe that the porcine auto-transplantation can serve as a representative model to predict the outcome of transplantation of human kidneys after multi-day perfusion.

This discrepancy/ translational gap between preclinical animal studies and (pre)clinical human studies has recently been acknowledged in a systematic review on animal models for kidney transplantation and ischemia reperfusion injury (10). Consequently, we believe that the pursuit of clinical translation for this perfusion platform should be directed elsewhere. We propose two alternative options to advance towards clinical application.

First, if animal studies are not feasible, one option is to transplant the perfused kidneys in brain-dead subjects and assess their function over a short period. A framework addressing the ethical, legal and social considerations for transplanting organs into brain-dead recipients has been previously established (11). This method has been accepted to advance xenotransplantation, and xenograft kidneys have been successfully transplanted into brain dead recipients with follow-up periods ranging from days to weeks (12, 13).

Alternatively, or in parallel, we propose to initiate a clinical safety and feasibility study employing a step-wise approach. This may entail an initial phase with 6-12 hours of perfusion, and upon successful demonstration of safety, progressing to extended durations, such as 24 hours and beyond. This approach is currently also being employed to evaluate the safety and feasibility of normothermic kidney perfusion (ISRCTN13292277 and NCT04693325).

The outcomes of these porcine kidney perfusions and transplantations have been added to the results section (line 267-294), and our considerations, as well as the observed limitations to the porcine kidney auto-transplantation model have been added to the discussion (line 356 to 372).

3. Perfusion parameters were stable, but it appears that even over the first 4 days there is evidence of the accumulation of damage. Increasing levels of lactate, KIM-1 and NGAL. The scale bars in figure 6 are deceiving for values of KIM-1 and NGAL. The levels of KIM-1 rise to approximately 4ng/mg which is considered high.

Response: We acknowledge the reviewers comment regarding the clarity of these graphs and have therefore modified the scale of the Y-axis for Figure 6H-J (KIM1, NGAL, respectively) (line 722-742).

4. What was the level of cold ischaemic in the discard kidneys. Two of the kidneys used for the 8 day ex vivo perfusion have significant kidney dysfunction at the time of donation. Can the authors comment? Was this injury apparent during perfusion?

Response: We would like to thank the reviewer for this question. We had not previously made a comparison between pre-existing injury at the time of donation and kidney specific markers of injury during multi-day perfusion. From the figure below it becomes apparent that the two kidneys with more pre-existing injury (Day8_1 and Day8_2), also released more injury markers during multi-day perfusion. This was most clear for urine KIM1. In clinical practice, there is unfortunately always uncertainty as to the level of ischemic injury in DCD donors which is exactly the reason that prolonged organ perfusion and the ability to assess function prior to transplantation would be so useful, similarly to what has been demonstrated in liver donation.

Figure. Kidney specific markers of injury during multi-day perfusion in urine and perfusate in relation to pre-existent kidney injury in the donor. The donors of kidneys 7-8 (Day8_1 and Day8_2) had more elevated levels of serum creatinine, as compared to the donors of kidneys 1-6 (Day8_3, and Day4_1-Day4_5). See Table S2 for donor background. **A-D**, Urine and perfusate levels of KIM1 and NGAL. **E-F**, Area under the curve (AUC) for urine and perfusate KIM1 and NGAL, respectively. Unpaired t tests with a False Discovery Rate of 1% were performed. Symbols: * $P \leq 0.05$, ** $P \leq 0.01$, *** $P \leq 0.001$.

Minor comments

1. Kidneys gained weight but with no evidence of oedema. The authors speculate that this indicated the vascular barrier function was preserved. Simply weighing the kidney is not an accurate measure. Wet/dry ratio would be more exact.

Response: We agree with the reviewer that weight gain alone does not represent the most accurate measure for vascular barrier integrity and have therefore removed this passage from the text (line 123-125). However, to be able to provide an accurate measure for vascular barrier function, labelled dextrans were added to our 4-days perfusion experiments (Figure 6J-K), which directly proofs intactness of the vascular barrier function throughout the first days of perfusion.

2. The group in Oxford have shown stable perfusion with red cells for up to 48h. This should be included in the discussion.

Response: We have included this into the discussion (line 337-338), referring to a recent publication describing 48h NMP of discarded human kidneys (14). However, we find it important to note that the authors of this manuscript concluded by stating that “*long-term, longer than 24 h, ex vivo perfusion of the kidney, however, might be limited by accumulating tubular damage as well as de-novo glomerular thrombotic events when currently available devices (clinically licensed for 6 h kidney NMP only) are applied*”. Within our manuscript we have demonstrated that, when perfused with a cell-free nutrient-supplemented perfusate at subnormothermia (25°C), perfusion up to 4 days is currently feasible.

Reviewer #2 (Remarks to the Author):

I read with interest the manuscript: NCOMMS-23-12780A-Z intitled: A cell-free nutrient-supplemented perfusate allows four-day ex vivo metabolic preservation of human kidneys.

This work is of obvious interest, with a first-rate technological approach. The aim is to infuse human discarded kidneys over several days using a home-made infusion circuit. In this study, the perfused human kidneys discarded for transplantation with a cell-free perfusate supplemented with TCA cycle intermediates at sub normothermia (25°C) up to 8 days.

The primary objective of this study was the development of a perfusion loop that supports multi-day ex vivo preservation of metabolically active human kidneys. The authors have followed a very demanding set of specifications for a model capable of meeting the constraints of organ perfusion and fulfilling their ambition.

Response: We highly appreciate the encouraging words of the reviewer. Our approach has been to continue the evaluation of kidneys after “loss of viability”, to the point of organ failure. We sought to describe what happens to a kidney as it fails during long-term perfusion. To this end, we employed our recently established single cell level dynamic metabolomics platform.

The methodology uses renal tissue recovered during perfusion for metabolic and functional analysis. Krebs cycle metabolites were added to study metabolic activity. The perfusion medium is described, with particular reference to the metabolites involved (glutamine, citrate and acetate). This prototype perfusion circuit Kidney perfusion and the parameters characterizing it are presented with their evolution during the 8 days and highlight modifications from 4 days onwards in terms of performance. The monitoring of lesion markers also shows a progressive evolution with a potential correlation with histological studies. I am concerned by 30% kidney weight gain (Day0: 283±39, Day8: 363±52 gram) which is intriguing and difficult to explain despite comments minimizing this weight gain.

Response: We agree with the reviewer that weight gain is an important parameter and have removed our comments minimizing the observed weight gain (line 123-125). Actual human kidney transplantation is probably required to provide insight in any true clinical significance of these perfusion parameters. At this point, however, we feel confident that this weight gain as observed in the human kidneys, although present, should not lead to any complications further on in the translation process, for several reasons:

- Previous research conducted on human kidney grafts showed that weight gain around 30% during (hypothermic) perfusion did not negatively impact transplantation outcomes (15).
- We believe that major concerns regarding weight gain reside in the question whether or not the vasculature remains intact during the course of perfusion. To this end, we added fluorescently labeled dextrans (Figure 6J-K), thereby providing evidence for a preserved vascular barrier during the human kidney four-day perfusions. Large molecular weight dextrans (500 kDa) were retained within the vascular compartment, whereas small molecular weight dextrans (10 kDa) were filtered into the urine.
- As elaborately described in paragraph 2.6 of the revised manuscript, the perfusion dynamics of porcine kidneys show remarkable differences as compared to the human kidney

perfusions. During only 4 days of porcine kidney perfusion, organ weight increased substantially (114-308%) with clear evidence of edema formation (Figure S6-7). Here, disruption of the vascular barrier seems undisputable. Therefore, in contrast to these observations, we believe that the vascular barrier function remains intact throughout the human kidney perfusions.

In addition to these viewpoints regarding the interpretation of the observed weight differences, we find it remarkable that weight gain and edema formation following human kidney perfusion are hardly ever mentioned in the literature on human kidney perfusion. The absence of published findings on this limits our ability to extrapolate our observations to previously published data in terms of clinical significance.

As far as donors are concerned, there is considerable variability in terms of cause of death and different clinical characteristics, which must be taken into account. LDH and creatinine levels are heterogeneous. In this context, the GFR estimation may not be relevant taking into account the period.

Assessments by the various techniques (MALDI-MSI measurement, Post-MALDI-MSI staining, Untargeted lipidomics) provide highly innovative and functional information of great importance.

Mass Spectrometry Imaging (MSI) is a surface mapping technology that allows the spatial information and relative abundance of analytes (proteins, peptides, lipids or small metabolites) to be determined directly from the surface of a biological sample section. Following MALDI-MSI measurements, we performed Uniform Manifold Approximation and Projection (UMAP) analysis based on the lipidomic data, combined with post-MALDI-MSI immunofluorescence staining for cell type identification. Using this method, the authors were able to distinguish different cell types and their evolution during perfusion a during perfusion, with stability between D4 and D6. Unsupervised hierarchical clustering of the different timepoints for each cell population revealed that lipid remodelling predominantly occurred between Day6 and Day8 of perfusion. From Day6 of perfusion onwards, the authors observed a progressive increase in the perfusate levels of malondialdehyde (MDA), an indicator of oxidative stress. The results highlight the disruption of the pro-oxidant balance, with the depletion of anti-oxidant defences. The spatial distribution of these oxidized phospholipids was assessed within the MALDI-MSI analysed biopsies and results supports that oxidized lipid species are absent between Day0 and Day4-Day6 of ex vivo perfusion, with progressive oxidative stress taking place between Day4-Day6 and Day8 of perfusion resulting in the accumulation of peroxidised lipid species towards Day8. These results lead to two comments: 1. this D4-D6 window needs to be clarified and specified in terms of impact, and 2. this oxidative stress needs to be analyzed in terms of its longer-term consequences.

Response: We fully concede with the reviewer that the apparent increase in oxidative stress and the subsequent lipid peroxidation after Day4 is something that deserves more clarification in terms of impact. The main reason for us to perfuse the human kidneys up to 8 days was to be able to shed light on potential culprits during long-term ex-vivo kidney perfusion. For now, this 4-day window with the preserved oxidative balance already impacts the field of prolonged ex-vivo perfusion and creates opportunities for many advanced applications. In our opinion, the accumulation of oxidative

stress and lipid peroxidation after several days of perfusion rather directs towards investigating their potential as biomarkers within a clinical framework. Assessment of the presence of these oxidized lipids or levels of anti-oxidant capacity on clinical biopsies, such as available from the QUOD biobank, could unravel potential predictors for long-term transplantation outcomes.

Metabolic analysis show that all renal cell types identified in the biopsies maintained active cell metabolism and could use glucose and glutamine as nutrient sources for glycolysis and the TCA cycle after 4- and 8-days of perfusion. It was suggested an active glycolysis pathway and the contribution of glucose to the TCA cycle until Day8 of perfusion and a higher contribution of glutamine as carbon source for the TCA cycle as compared to glucose. Finally, conclusion from authors is that their finding underpins that sub normothermic machine perfusion allows multi-day ex vivo metabolic preservation of diseased human donor kidneys.

On the subject of renal function assessment, the existence of diuresis is not a rigorous means of assessing renal function. Tubular functions are nevertheless modest (despite the optimistic comments), and there is significant natriuresis and glucosuria indicating poor functionality.

The results presented in this work provide very interesting data on the metabolic state of the organs perfused by the system described.

On the other hand, the limitations described for the porcine model, and especially the conditions described with a 3h reperfusion, are fully justified. It would probably have been appropriate to propose a follow-up of one to two weeks.

Response: We thank the reviewer for their critical appraisal of the porcine kidney transplantation we previously described. To better address whether porcine transplantation would have been a suitable step towards clinical translation of our platform we set up a novel collaboration with the group of dr. Markus Selzner in Toronto, where they have a well-established porcine kidney auto-transplantation model (7-9).

However, what we found is that the perfusion dynamics of 3-month-old male porcine living-donor kidneys differed considerably from our previous observations in human kidneys (*Figure S6-7, and below*). During porcine kidney perfusions, vascular resistance began to rise between 12-24 hours, leading to a gradual reduction in renal flow. Furthermore, at the conclusion of perfusion, the porcine kidneys had gained 114-308% compared to their initial weight, a notable contrast to our findings in human kidneys (*Figure S6, and below*).

Figure. Comparison of perfusion dynamics between porcine kidneys and human kidneys. A, Vascular resistance. B, Renal flow. C, Whole organ weight at start and end of perfusion. D, Weight change during perfusion. E, Vascular resistance dynamics in porcine and human kidneys. F, Renal flow dynamics in porcine and human kidneys. For E and F * denotes the termination of porcine kidney perfusion #2. G, Weight change during porcine and human kidney perfusion.

Despite the variations between these porcine kidney perfusions and the previously perfused human kidneys, both Kidney#1 and Kidney#3 underwent auto-transplantation. Kidney#1 was transplanted following a contralateral nephrectomy. Initially, a uniformly perfused pink kidney was observed after reperfusion. However, in the following hour the kidney developed haemorrhagic infarction (see below) and the pig entered a distributive shock that required large doses of pressors and fluids. We believe the intraparenchymal blood was a reflection of disturbed vascular permeability, as was also supported by the extraordinary weight gain of the kidney during perfusion preservation.

For the transplantation of Kidney#3 the contralateral kidney remained in situ. After reperfusion, a patchy appearance with pink and darker regions was noted, as is often observed in DCD donations. Reperfusion differed from Kidney#1, as minimal pressors and fluids were required to maintain blood pressure. The pig was sacrificed after 7 days. Again, the kidney showed haemorrhagic infarction and intraparenchymal blood, reflective of disrupted vascular permeability, albeit to a lesser extent when compared to kidney#1.

Figure. Macroscopic appearance of the porcine kidneys at different timepoints.

It should be emphasized that such loss of capillary permeability was not noted during the human kidney perfusions. For example, the sieving coefficient for dextrans was maintained and weight gain was modest. Therefore, given these distinct differences in perfusion dynamics, we do not believe that the porcine auto-transplantation can serve as a representative model to predict the outcome of transplantation of human kidneys after multi-day perfusion.

This discrepancy/ translational gap between preclinical animal studies and (pre)clinical human studies has recently been acknowledged in a systematic review on animal models for kidney transplantation and ischemia reperfusion injury (10). Consequently, we believe that the pursuit of clinical translation for this perfusion platform should be directed elsewhere. We propose two alternative options to advance towards clinical application.

First, if animal studies are not feasible, one option is to transplant the perfused kidneys in brain-dead subjects and assess their function over a short period. A framework addressing the ethical, legal and social considerations for transplanting organs into brain-dead recipients has been previously established (11). This method has been accepted to advance xenotransplantation, and xenograft kidneys have been successfully transplanted into brain dead recipients with follow-up periods ranging from days to weeks (12, 13).

Alternatively, or in parallel, we propose to initiate a clinical safety and feasibility study employing a step-wise approach. This may entail an initial phase with 6-12 hours of perfusion, and upon successful demonstration of safety, progressing to extended durations, such as 24 hours and beyond. This approach is currently also being employed to evaluate the safety and feasibility of normothermic kidney perfusion (ISRCTN13292277 and NCT04693325).

The outcomes of these porcine kidney perfusions and transplantations have been added to the results section (line 267-294), and our considerations, as well as the observed limitations to the porcine kidney auto-transplantation model have been added to the discussion (line 356 to 372).

Furthermore, the use of two different preservation solutions for human and porcine kidneys is not clearly justified and would warrant further clarification.

Response: We would like to apologize to the reviewer for any unclarity. The same perfusate was used for the initial porcine perfusions and subsequent human perfusions. There is, however, a difference between the perfusate used and the substitution solution. The perfusate is what is present within the perfusion circuit. The substitution solution is what is continually added to the perfusate by the dialysis machine, whilst perfusate is removed from the circuit over a dialysis filter. The major difference between the perfusate and substitution solution is the absence of albumin in the substitution solution, as this is retained by the dialysis filter.

It would have been useful to present the prospects for the human clinic and how this platform could be a precursor for evaluation and repair/regeneration structures (HUB concept).

Response: We thank the reviewer for their suggestion and have included our view on the platform's clinical application to the discussion (see below, and line 374-382).

“By preserving kidneys in a metabolically active state for days rather than hours, we open possibilities for further advancements in transplantation. Existing protocols already direct donor organs through specialized facilities known as Organ Perfusion and Regeneration (OPR) units for assessment and potential pre-treatment using short periods of (cold) machine perfusion. Remarkable breakthroughs in liver preservation highlight the vast potential for prolonged graft preservation, such as the successful transplantation of previously discarded livers (44), transplantation after 3-day ex vivo preservation (9), immuno-modulation during machine perfusion (10), and even bile duct regeneration through cholangiocyte organoid transplantation (45). These advances underscore the transformative impact of prolonged organ preservation and demonstrate the game-changing potential in the field of transplantation.”

Minor comment: is substitution the correct word ?

Response: We would like to thank the reviewer for noting this. Indeed, it should have said substitution solution instead of substitution.

Reviewer #3 (Remarks to the Author):

De Haan et al: A cell-free nutrient-supplemented perfusate allows for 4 day ex vivo metabolic preservation of human kidneys

The authors performed mass spectrometry imaging analyses in the context of human kidney perfusion ex vivo. The authors combine ex vivo culture, analyses of biopsies with physiological measurements and chiefly with an imaging MS-platform. The use of isotopes is an elegant strategy used here, yet the authors do not demonstrate supporting data that their quantification is sound. In total, the identification of metabolites and their quantification raises some concerns.

The main concern is that the findings the authors present are not corroborated by other methods. Presented as is, conclusions cannot be made regarding the nature of metabolic changes, and appear to be made by a very selected observation of only a few cells. Bulk analyses with additional LC separations are necessary to confidently assign identities to the claimed metabolites and their isotopologues. A global overview of quantification of fluxes is missing. Code and raw data are not provided. The authors provide overall cherry-picked metabolic pathways, chiefly focusing on use of glucose, but not other substrates such as fatty acids, or lactate as fuels.

Response: We thank Reviewer#3 for their critical appraisal of our manuscript and the provided feedback. Within this manuscript we applied our spatially resolved single cell resolution isotope tracing platform in the context of ex vivo human kidney perfusion. In previous work we validated this spatial isotope tracing approach, using it to analyze cell type specific dynamics of metabolism in kidney repair following bilateral ischemia reperfusion injury in rodents (16). Since, we have extended it to define metabolic cell fate trajectories in human kidney differentiation (17). Now, we have applied the same approach, to study cell type specific dynamics of the adult human kidney during ex vivo preservation.

Major concerns:

1. The high resolution MS imaging approach is generally elegant, and 20ppm should be sufficient to resolve those peaks. However, spectral quality and S/N ratios need to be good enough in order to really make the conclusions the authors make. Aspartate has a mass of 132.02968267, and malate has one of 132.006973 – can the authors show that these two metabolites are really distinguishable without LC? Similar, Glutamate has a mass of 146.04533274, glutamine 146.06914219 – are the spectral data clean enough to demonstrate the different metabolic behavior of these two metabolites?

Response: We used NEDC matrix in negative mode, this generally leads to singly charged negative ions. This fact translates into the following observed m/z values for the listed metabolites: aspartate m/z 132.0302; malate (singly charged) m/z 133.0142; glutamine is m/z 145.0619 and glutamate is m/z 146.0459. In turn, there is a 1Da difference between these analytes rendering distinction easily possible. Of note, in mass spectrometry a mass to charge ratio (m/z) is being analyzed, hence masses of the same charge state need to be compared.

This is particularly concerning when labels are very low, and noise is likely high i.e. less than 1 % as in Fig. 5N, F.

Response:

The majority of the labels are higher than 1% in Figure 5F and N. We have applied a cut-off of $s/n > 3$, when selecting the labelled peaks. Considering the heterogeneity of the tissue it is normal to observe some pixels with a low labelling percentage.

This is in line with observations by another group that assessed spatial isotope tracing in rodent kidneys, published in Nature Methods (see figure below) (18).

[FIGURE REDACTED]

Figure. Kidney isotope labelling patterns (Figure 3A and C from Wang et al. 2022. Nature Methods (18)) Colors reflect fractional abundance of the indicated isotopic forms.

Another concern is the measurement of glutathione (Figure 4D): GSH is so challenging to quantify since it undergoes autooxidation, how are the authors sure that this is really this metabolite?

Response: This is an important remark that we would like to address as follows: First, the reviewer is right, analyzing GSH can be cumbersome. However, this is mainly caused by necessary sample processing (i.e. storage, extraction, etc). In our case the investigated material was flash frozen and underwent analysis swiftly after thawing thus limiting autooxidation of GSH and making it accessible to MSI analysis.

Nevertheless, to further strengthen this assumption we performed MS/MS analysis on human kidney tissue using a recently installed TimsTof MALDI2 system at different collision energy voltages to induce GSH fragmentation. As can be seen below, in the m/z 306.0769 region only one major peak was found, which corresponds to the m/z value of GSH (ppm = 1). MS/MS analysis and comparison of CID spectra with a genuine standard presented in the literature further confirmed the identity being GSH. We have added this information to the method section (line 517-519).

Figure. In situ MS/MS analysis for the identification of GSH. A, In situ MS/MS measurement on human kidney tissue using TimsTof MALDI2 with 0 eV collision energy. **B,** In situ MS/MS analysis on human kidney tissue using TimsTof MALDI2 with 20 eV collision energy. **C,** MS/MS fragmentation of GSH in the ESI negative mode from a previous report (19).

Editorial note: Additional citation for panel c. "Reprinted (adapted) with permission from Chem. Res. Toxicol. 2005, 18, 4, 630–638 Publication Date: March 9, 2005 <https://doi.org/10.1021/tx049741u> Copyright © 2005 American Chemical Society. Copyright 2005 American Chemical Society."

2. Figure 2: How many images were taken into account for the quantification? I thought that it is expected that the tissue heterogeneity is very high even macroscopically, so several images per kidney are likely necessary? How are the authors sure that their "PT clusters" are not a simple result of over-clustering? Based on how many cells have these analyses been done? It seems the amount of cells is rather limited. The overall area imaged needs to be depicted, not just the number of kidneys.

Response: Tissue punch biopsies (4mm in diameter) were taken at different time points during 8-day organ culture (Day0, Day2, Day4, Day6, Day8). At each timepoint, one biopsy was taken and cut longitudinally into two pieces of which one was fixed in paraformaldehyde and the other snap frozen in liquid nitrogen. On Day0, Day4 and Day8 an additional biopsy was taken for ¹³C-labelling experiments. Taking additional biopsies would have increased the chance of compromising perfusion hemodynamics, as biopsies often cause some perfusate leakage, and there with local pressure changes.

The amount of tissue assessed per timepoint is comparable to clinical practices, where renal biopsies are sometimes needed to make diagnosis. Per biopsy one region of the cortex was imaged and used for subsequent analysis. Although some biopsy-location dependent heterogeneity may be present, heterogeneity between different donor kidneys should be bigger. For the different kidneys we observed the same changes in lipid species distribution and abundance over time (Figure 3), making it very unlikely that biopsy-location derived heterogeneity biased the results.

Based on both a bioinformatics perspective, as a biological perspective, it seems very unlikely that the PT clusters are the result of over-clustering. Figure 2D displays the cluster-specific lipid features for each PT cluster identified, demonstrating clear differentiation between each cluster. Within the renal cortex there are two different types of proximal tubules, namely PT-S1 and PT-S2. It is very likely that PT1 and PT2 are corresponding to these proximal tubule phenotypes, as they form the fast majority on Day0 (Figure 2I). However, it has not been our aim to identify the phenotype of the observed PT1 and PT2 cluster in this manuscript. More importantly, the PT3 cluster only starts to appear during perfusion (Figure 3; *m/z* 837.5 and *m/z* 865.6 are barely present at the start of perfusion), indicating an abnormal PT phenotype that is related to the oxidized lipids (Figure 4).

In total, 33 sections were assessed through MALDI-MSI, comprising a total of 2.394.033 pixels analysed for this study, resulting in an average of 72.546 pixels per section at a spatial resolution of 5 x 5 μm^2 . The average measured area for one section was 1.8 mm^2 . Proximal tubules constitute approximately 60% of the kidney cortex area, ensuring a sufficiently large population of proximal tubular cells. We have added this information to the methods section (line 493-495).

3. All data are quantified as $n=1$ without SD/SEM and presented with one sample t-test which is not appropriate if three replicates were analyzed. In Fig3: \log_2 fold changes need to be added.

Response: We agree with the reviewer's observation regarding the one sample t-test. Given the variability among different donors and the MSI measurements across batches, we opted to present the relative fold change compared to the T0 sample from the same kidney. Considering the sample size ($n=3$), no suitable statistical test could be applied, leading us to remove the one-sample t-test and instead only show the trend of trend of changes in Figures 3 and 4.

4. Information from the untargeted lipidomics and the MS imaging should be integrated better. Lists/excel files of identified metabolites/features and their kinetics are not provided. The untargeted analysis of lipids could be better described (MS DIAL) – but how exactly were lipids identified and annotated is unclear.

Response: A more detailed description on how MS-DIAL was used for the identification of lipids has been added to the methods section of the manuscript (line 618-628). In brief, all peaks with an intensity of at least 200 ions eluting between 0.5 and 10 min of the chromatogram were considered. Alignment within samples was performed with a retention time and MS1 tolerance of 0.15 min and 0.025 Da, respectively. MS-DIAL lipid database version Msp20221205132019 was used for lipid annotation having the following parameters as identification settings: 0.01 and 0.05 Da for both MS1 and MS2 accurate mass tolerance, respectively, and including [M-H]⁻, [M-H₂O-H]⁻ and [M+formic acid-H]⁻ as adducts. Manual curation for the oxPLs species was performed by only including those oxPLs for which the experimental and reference MS/MS spectra matched. The annotation of any other lipid class has not yet been verified as this was out of the scope of this work. Subsequently, matching of untargeted lipidomics data with the results obtained from MALDI-TOF and MALDI-FTICR MSI analysis was based on matching accurate masses (5 ppm) of consistently observed oxPL.

We have included a list of annotated oxPLs as source source data on Figshare. This has been deposited on FigShare and can be accessed through the following link:

<https://figshare.com/s/aa255566ca4f6a2a0702>.

5. I cannot review their bioinformatics pipeline, since no information on bioinformatics is provided. Full data and code transparency is necessary to provide a review of the paper.

Response: We concur with the reviewer's remarks concerning data and code transparency. In this study, we used the same code as in our previously published articles (16, 17), for which the code has been deposited in Github (<https://github.com/Ganggiwang/scDYMO>). The code has been validated during previous peer-review.

The raw MALDI-MSI data reported in this study cannot be deposited in a public repository because of the large data size (about 2 TB). Similar with our previous publications, we will deposit all the exported and processed data in FigShare upon acceptance of our manuscript. We will also provide the raw data to those who are interested through data transfer upon reasonable request.

The pre-processed and processed MALDI-MSI data can be accessed through the following links:

Pre-processed data: <https://figshare.com/s/8a7acd4f942326ca57c9>

Processed data: <https://figshare.com/s/e5c9d73bfa7bfe9a1c7a>

6. In Fig. 5 scale bars are vastly different between the individual replicates, giving a false impression of the real signals and isotope ratios. Can the authors apply the same scale bar to all images?

Response: We understand the reviewer's concern regarding the scale bars. The purpose of the images in Figure 5 was to demonstrate spatial heterogeneity within the tissue at the different timepoints of ex vivo preservation (Day0, Day4, Day8). If we apply the same scale bar to all the images, there will be less detail in displaying heterogeneity due to the differences between samples (see an example in the image below). In addition, all images are accompanied by graphs which show the differences between the samples.

Figure. Images displaying the spatial glutamate M+5 enrichment after 2 hours of incubation with U-¹³C₅-glutamine incubation in biopsies taken at Day0, Day4 and Day8, respectively.

7. PT1,2,3 are behaving very similarly metabolically (Fig. 5) – is this expected? Is it then necessary to distinguish them at all? It seems like the authors do not take advantage of their spatial resolution by looking at other cells such as glomeruli – why not? Again, a larger aggregation of data – instead of looking at 1 panel at a time – would be useful.

Response: As correctly pointed out by the reviewer, this MSI approach also allows us to look at other cell types in the tissue as well (this information is in the data set, see Figure 2). However, we decided to focus on metabolic preservation of the proximal tubule population because of the following reasons: The proximal tubules have the highest metabolic activity within the kidney, requiring substantial energy production by mitochondria to maintain renal function (a.o. tubular transport machinery). It is thus the “Achilles heel” of ischemic kidney injury. Consequently, PT-cells are particularly susceptible to damage following ischemia reperfusion injury (20). Therefore, we focused our current analysis on the metabolic dynamics within the different PT clusters identified, as opposed to other cell populations.

As mentioned above, we identified three PT-cell clusters based on cluster-specific lipid features. Of these clusters, PT1 and PT2 likely correspond to the two types of proximal tubular cells naturally present within the cortex, namely PT-S1 and PT-S2. More importantly, the PT3 cluster only became apparent during perfusion, indicating an abnormal phenotype that is related to the oxidized lipids. While PT1,2,3, reflect different phenotypes based on their lipidome profile they are still within the same anatomical proximal tubule compartment, serving the same functions such as mass solute reabsorption. From that perspective it is not strange that they may not differ in processes such as glycolysis and TCA activity.

8. I am not an expert in ex vivo transplant perfusion, but it seems to me that a “viability in cell culture” /metabolic fitness aspects needs more data to confirm that these kidneys are viable and respiratory capacity is conserved. Why did the authors not elect to use the gold standard for metabolic capabilities, i.e. seahorse assay?

Response: Our assessment of kidney perfusion dynamics also included measurements of oxygen consumption, pH, lactate release, and glucose uptake, albeit at the whole organ level (Figure 1, Figure 6). Furthermore, we applied MSI together with isotope tracing to study dynamic metabolomic changes in nutrient partitioning over time, information that the Seahorse assay does not provide.

From a practical standpoint, conducting these perfusions involves a substantial workload. The unpredictable nature of receiving human kidneys deemed unsuitable for transplantation, occurring at any time of the week, including weekends, and during both day and night, poses a significant challenge. Coordinating both the perfusions and ^{13}C -incubations simultaneously has proven to be a considerable logistical hurdle. Including an additional assay, such as SeaHorse, which necessitates real-time execution, would not have been feasible.

9. The authors use Q-Flux equations to determine fluxes based on Hubbard et al. Cell Metabolism 2023. These, however, are based on tracer steady state infusion of glutamine in mice. This was probably not the case here. The authors should provide more validation that this approach is really applicable.

Response: Indeed the Q-flux equations are based on tracer steady state. However, in Extended Data Figure 4 of our previous publication (16), we showed that 2 hour *ex vivo* incubation with ^{13}C -glutamine is sufficient to reach a pseudo-steady state for TCA-cycle metabolites (see figure 1 below). We estimated the fractional ^{13}C enrichment of glutamate (Glu M+5) and malate (Mal M+4) to be 20% and 16%, respectively, at 2 h incubation.

Similar ranges of ^{13}C enrichment (around 20% and 16% after normalization to serum enrichment) were also found in kidney tissue in an 150 min *in vivo* infusion experiment (see figure 2 below; source: Wang et al. Nature Methods 2022 (18)) in which the authors showed a pseudo steady state of the kidney. Thus, we can support that our method of 2 hours *ex vivo* incubation of ^{13}C -glutamine is sufficient to assume a pseudo-steady state for the TCA cycle metabolites presented here. We have added this to the results section (line 234-235) and methods (line 572-574)

Figure 1. Pseudo-steady state is reached within 2 hours of *ex vivo* incubation with $\text{U-}^{13}\text{C}_5$ -glutamine (E) and $\text{U-}^{13}\text{C}_{18}$ -linoleate (F). Graphs showing the curve of ^{13}C enrichment of isotopologues of metabolites over a time course (up to 2 h). (Source: Extended Data Figure 4 from our previous publication Wang et al. 2022. Nature Metabolism (16)).

[FIGURE REDACTED]

Figure 2. Pseudo-steady state after 150 minutes of *in vivo* infusion of $\text{U-}^{13}\text{C}_5$ -glutamine. Absolute labelling of the indicated metabolites from $\text{U-}^{13}\text{C}_5$ -glutamine tracer. Colors reflect the fractional abundance of the indicated isotopic forms (without normalization to serum

tracer labelling, which is shown as a bar graph). (Source: Wang et al. Nature Methods 2022; (18))

Line 468: Figure 3K does not exist.

Response: Thank you for noting this. It should have stated Figure 2K.

References

1. Lau N-S, Ly M, Dennis C, Jacques A, Cabanes-Creus M, Toomath S, et al. Long-term ex situ normothermic perfusion of human split livers for more than 1 week. *Nature Communications*. 2023;14(1):4755.
2. Zulpaite R, Miknevičius P, Leber B, Strupas K, Stiegler P, Schemmer P. Ex-vivo Kidney Machine Perfusion: Therapeutic Potential. *Front Med (Lausanne)*. 2021;8:808719.
3. Hendriks KDW, Brüggewirth IMA, Maassen H, Gerding A, Bakker B, Porte RJ, et al. Renal temperature reduction progressively favors mitochondrial ROS production over respiration in hypothermic kidney preservation. *J Transl Med*. 2019;17(1):265.
4. Bruinsma BG, Yeh H, Ozer S, Martins PN, Farmer A, Wu W, et al. Subnormothermic machine perfusion for ex vivo preservation and recovery of the human liver for transplantation. *Am J Transplant*. 2014;14(6):1400-9.
5. Zarnitz L, Doorschodt BM, Ernst L, Hosseinejad A, Edgworth E, Fechter T, et al. Taurine as Antioxidant in a Novel Cell- and Oxygen Carrier-Free Perfusate for Normothermic Machine Perfusion of Porcine Kidneys. *Antioxidants (Basel)*. 2023;12(3).
6. von Horn C, Zlatev H, Lüer B, Malkus L, Ting S, Minor T. The impact of oxygen supply and erythrocytes during normothermic kidney perfusion. *Scientific Reports*. 2023;13(1):2021.
7. Kathis JM, Echeverri J, Goldaracena N, Louis KS, Yip P, John R, et al. Heterotopic Renal Autotransplantation in a Porcine Model: A Step-by-Step Protocol. *J Vis Exp*. 2016(108):53765.
8. Parmentier C, Gao F, Ray S, Kawamura M, Noguiera E, Ganesh S, et al. Intubation, Central Venous Catheter, and Arterial Line Placement in Swine for Translational Research in Abdominal Transplantation Surgery. *J Vis Exp*. 2023(192).
9. Kathis JM, Echeverri J, Linares I, Cen JY, Ganesh S, Hamar M, et al. Normothermic Ex Vivo Kidney Perfusion Following Static Cold Storage-Brief, Intermediate, or Prolonged Perfusion for Optimal Renal Graft Reconditioning? *Am J Transplant*. 2017;17(10):2580-90.
10. Lerink LJS, de Kok MJC, Mulvey JF, Le Dévédec SE, Markovski AA, Wüst RCI, et al. Preclinical models versus clinical renal ischemia reperfusion injury: A systematic review based on metabolic signatures. *Am J Transplant*. 2022;22(2):344-70.
11. Parent B, Gelb B, Latham S, Lewis A, Kimberly LL, Caplan AL. The ethics of testing and research of manufactured organs on brain-dead/recently deceased subjects. *J Med Ethics*. 2020;46(3):199-204.
12. Porrett PM, Orandi BJ, Kumar V, Houp J, Anderson D, Cozette Killian A, et al. First clinical-grade porcine kidney xenotransplant using a human decedent model. *Am J Transplant*. 2022;22(4):1037-53.
13. Montgomery RA, Stern JM, Lonze BE, Tatapudi VS, Mangiola M, Wu M, et al. Results of Two Cases of Pig-to-Human Kidney Xenotransplantation. *New England Journal of Medicine*. 2022;386(20):1889-98.
14. Messner F, Soleiman A, Öfner D, Neuwirt H, Schneeberger S, Weissenbacher A. 48 h Normothermic Machine Perfusion With Urine Recirculation for Discarded Human Kidney Grafts. *Transpl Int*. 2023;36:11804.
15. Wilson CH, Gok MA, Shenton BK, Balupuri S, Gupta AJ, Asher J, et al. Weight increase during machine perfusion may be an indicator of organ and in particular, vascular damage. *Ann Transplant*. 2004;9(2):31-2.
16. Wang G, Heijs B, Kostidis S, Mahfouz A, Rietjens RGJ, Bijkerk R, et al. Analyzing cell-type-specific dynamics of metabolism in kidney repair. *Nature Metabolism*. 2022;4(9):1109-18.
17. Wang G, Heijs B, Kostidis S, Rietjens RGJ, Koning M, Yuan L, et al. Spatial dynamic metabolomics identifies metabolic cell fate trajectories in human kidney differentiation. *Cell Stem Cell*. 2022;29(11):1580-93.e7.
18. Wang L, Xing X, Zeng X, Jackson SR, TeSlaa T, Al-Dalahmah O, et al. Spatially resolved isotope tracing reveals tissue metabolic activity. *Nat Methods*. 2022;19(2):223-30.

19. Dieckhaus CM, Fernández-Metzler CL, King R, Krolikowski PH, Baillie TA. Negative Ion Tandem Mass Spectrometry for the Detection of Glutathione Conjugates. *Chemical Research in Toxicology*. 2005;18(4):630-8.
20. Gerhardt LMS, Liu J, Koppitch K, Cippà PE, McMahon AP. Single-nuclear transcriptomics reveals diversity of proximal tubule cell states in a dynamic response to acute kidney injury. *Proc Natl Acad Sci U S A*. 2021;118(27).

REVIEWERS' COMMENTS

Reviewer #1 (Remarks to the Author):

I am satisfied with the changes to the manuscript and recommend to accept the manuscript

Reviewer #2 (Remarks to the Author):

I read with interest the revised version of the manuscript

#: NCOMMS-23-12780B

Title: A cell-free nutrient-supplemented perfusate allows four-day ex vivo metabolic preservation of human kidneys

The authors provided precise answers and clarifications.

I recommend publication of the manuscript

Reviewer #3 (Remarks to the Author):

The authors have responded to all my comments. The glutathione data should probably be removed, since the quantification (vs autooxidated form) is not reliable.

Reviewer #3 (Remarks on code availability):

It is disappointing to not see documented code for this paper.

RESPONSE TO REVIEWERS (2)

Reviewer #3 (Remarks to the Author):

The authors have responded to all my comments. The glutathione data should probably be removed, since the quantification (vs autooxidated form) is not reliable.

Response: We have removed the glutathione data from the manuscript (Figure 4).

Reviewer #3 (Remarks on code availability):

It is disappointing to not see documented code for this paper.

Response: The code that has been used for the analysis of the data underlying this manuscript has been previously published (<https://doi.org/10.1038/s42255-022-00615-8>) and is available at <https://doi.org/10.5281/zenodo.7191331>.